# Variable Responses to a Marine Heat Wave in Five Fringing Reefs of Southern Taiwan

**Zong-Min Ye [1], Anderson B. Mayfield [2,3,* and Tung-Yung Fan [1,4,*]**

1. National Museum of Marine Biology and Aquarium, Checheng, Pingtung 944, Taiwan; tony83111@gmail.com
2. International Coral Reef Society, Tavernier, FL 33070, USA
3. Coral Reef Diagnostics, Miami, FL 33129, USA
4. Department of Marine Biotechnology and Resources, National Sun Yat-Sen University, Kaohsiung 804, Taiwan
* Correspondence: anderson@coralreefdiagnostics.com (A.B.M.); tyfan@nmmba.gov.tw (T.-Y.F.);
  Tel.: +1-337-501-1976 (A.B.M.); +886-8-8825001 (ext. 2248) (T.-Y.F.)

**Abstract:** In 2020 marine heatwaves elicited severe bleaching on many of Earth's coral reefs. We compared coral reef benthic community composition before (April 2020), during (September 2020), and after (December 2020–September 2021) this event at five fringing reefs of Southern Taiwan. The four shallow (3 m) reefs were hard coral-dominated in April 2020 (cover = 37–55%), though non-bleached coral cover decreased to only 5–15% by December 2020. Coral abundance at the two shallow (3 m), natural reefs had failed to return to pre-bleaching levels by September 2021. In contrast, coral cover of two artificial reefs reached ~45–50% by this time, with only a small drop in diversity. This is despite the fact that one of these reefs, the Outlet, was characterized by temperatures >30 °C for over 80 days in a six-month period due not only to the bleaching event but also inundation with warm-water effluent from a nearby nuclear power plant. Only the lone deep (7 m) reef was spared from bleaching and maintained a coral/algal ratio >1 at all survey times; its coral cover actually increased over the 18-month monitoring period. These data suggest that (1) the natural deep reef could serve as a refuge from thermal impacts in Southern Taiwan, and (2) the remaining corals at the Outlet have either adapted or acclimatized to abnormally elevated temperatures.

**Keywords:** benthic ecology; conservation; coral bleaching; coral reefs; marine heatwaves; refugia; restoration; thermal effluent; upwelling

## 1. Introduction

Earth has fully entered the "Anthropocene," an era in which no ecosystem has been spared from the wide reach of civilization [1]. Coral reefs in particular have been severely degraded due overfishing, pollution, and climate change (especially the associated ocean warming [2]), and marine heatwaves will only become more common and extreme in years to come [3]. Coincidingly, mass coral bleaching events, whereby the coral–dinoflagellate (family Symbiodiniaceae) endosymbiosis breaks down, are predicted to increase in frequency and magnitude, with widespread coral loss expected to have immense impacts on marine ecosystem function and services [4]. This poses a substantial risk to the myriad marine organisms that depend on these ecosystems, as well as the livelihoods of those individuals that rely on coral reefs for sustenance and income (e.g., fisheries & tourism). High temperature-induced bleaching events have been well documented since the early 1980s, with global scale events occurring in 1998, 2010, 2014–2017, and 2020 [5]. The frequency and intensity of bleaching events are rapidly approaching unsustainable levels since, in the absence of acclimatization or adaptation, there is no longer sufficient recovery time between events [6]. Some corals can shift to heterotrophy in their bleached states [7], but this tends to be a short-term survival strategy at best; unless thermotolerant dinoflagellate endosymbionts are rapidly re-uptaken, bleached corals will perish [8].

The abiotic milieu of coral reef habitats (e.g., temperature, light, pH, flow, etc.) can change dramatically over short-term time-scales (e.g., the tidal cycle [9]) and differ markedly across small spatial gradients (~cm–m [10]); this extensive variation can modulate bleaching frequency and severity [11]. For example, corals of highly variable temperature environments have been shown to be "stress-hardened" to better combat future increases in temperature [12–15]. The type of reef, whether it be natural or "artificially modulated," defined herein as a habitat either shaped directly by human activities (e.g., a seawall) or one whose structural and/or abiotic environment has been fundamentally altered by human activities at the local scale (beyond the global climate change-induced changes experienced by all reefs), could also have implications for the response of the resident corals to temperature increases, especially since their community structures and thermal histories may fundamentally differ from those of natural reefs [16–19].

Fringing coral reefs are well-developed along the coast of Southern Taiwan's Hengchun Peninsula, and reefs within Taiwan's southernmost embayment, Nanwan Bay, frequently experience cold-water intrusions driven by internal wave-induced upwelling and mesoscale eddies [20]. One reef within Nanwan Bay, the "Outlet" (OL), is episodically immersed in the warm-water effluent from a nearby nuclear power plant; this thermally charged wastewater, in combination with an El Nino–Southern Oscillation (ENSO) event, caused the first documented bleaching events at OL in 1987 and 1988 [21]. Specifically, bleaching occurred at around four "degree-heating weeks" (DHWs) according to the National Oceanic and Atmospheric Administration's (NOAA) Coral Reef Watch (CRW). Since then, region-wide coral bleaching events have occurred in 1998, 2007, 2016, 2017, and 2020 (with minor ones in 2010 & 2014 [20]). The UN Environment Programme [22] predicts that bleaching will become an annual phenomenon in Southern Taiwan by 2037.

Coral abundance in Nanwan Bay has fluctuated over time as a result of these bleaching events [23], as well as annual typhoons that occur in the boreal summer [24]. The spatial heterogeneity in coral abundance and recovery [25] suggests that reef resilience could vary over small (m–km) spatial scales (*sensu* [26]). Nanwan Bay could consequently be an ideal region for understanding climate change impacts on coral reefs [27]. Therefore, we monitored seawater temperature and benthic community structure at three natural (including both upwelling & non-upwelling reefs) and two highly artificially modulated (hereafter "artificial") fringing coral reefs of Southern Taiwan (Figure 1) across the most recent of the aforementioned bleaching events (2020) in an effort to better understand the implications of seawater temperature rise on the local reef ecology and determine whether artificial reefs, most notably the OL, are now better acclimatized (or even adapted) to high seawater temperatures on account of their anthropogenically forced thermal history.

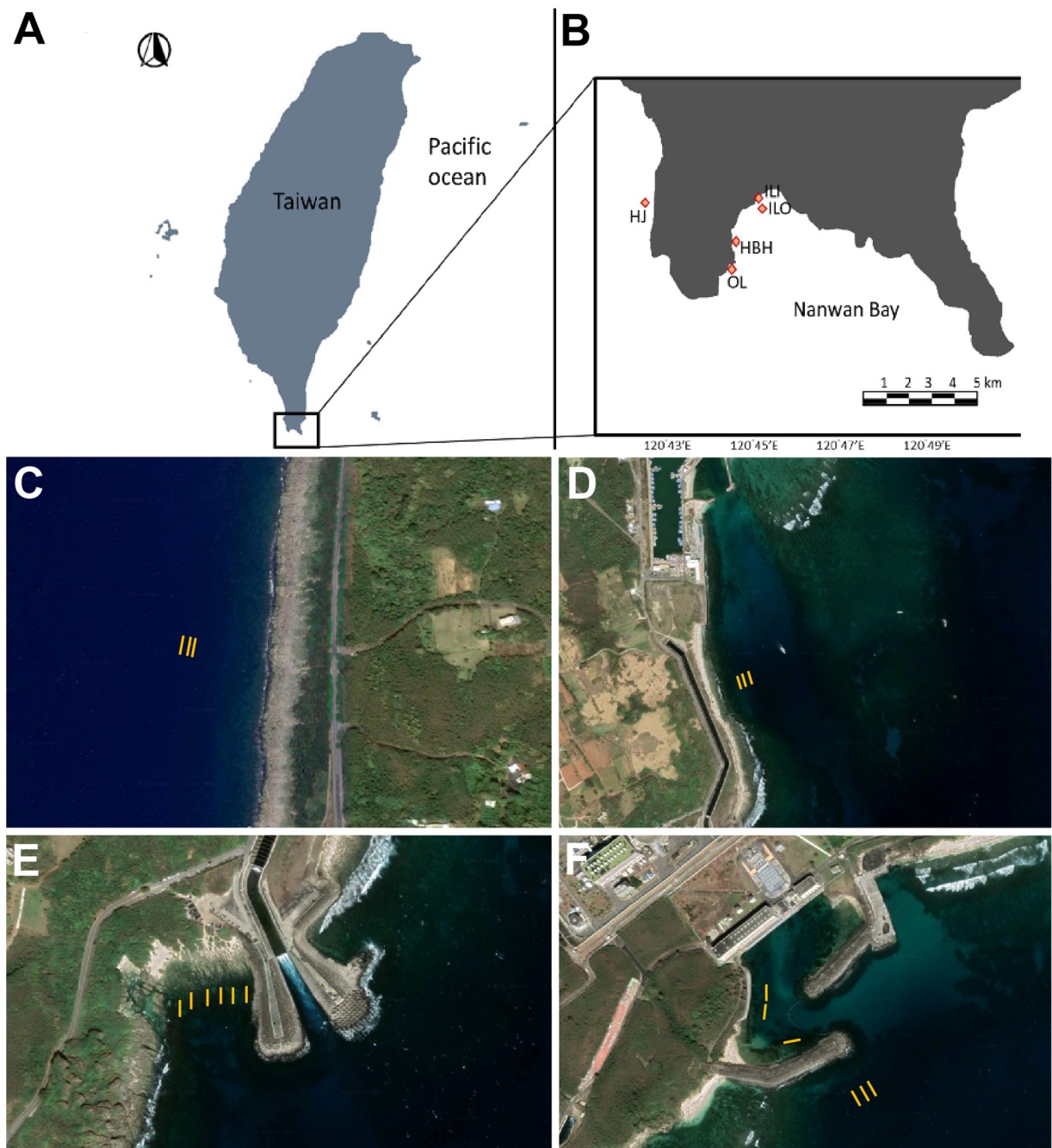

**Figure 1.** A map of Taiwan (**A**) with inset depicting the location of the study sites (**B**). Panels (**C–F**) show the shallow (3 m), natural, non-upwelling reef "Hejie" (HJ); the shallow, natural, upwelling reef "Houbihu" (HBH); the shallow, upwelling and nuclear power plant thermal effluent-influenced "Outlet" (OL) reef; and two reefs near the same power plant's intake water area ("inlet"), respectively. The latter two include an upwelling reef within an artificially constructed inlet, "Inlet-inside" (ILI), and a deeper (7 m), natural, upwelling reef offshore of the inlet, "Inlet-outside" (ILO). The yellow lines indicate transect locations (*n* = 3 for all sites except for OL [*n* = 5–6]).

## 2. Materials and Methods

### 2.1. Study Sites and Benthic Surveys

Seawater temperature profiles (Figure S1A–E) and benthic community structure were assessed at five Southern Taiwanese fringing reefs (Figure 1A,B) with very different characteristics (Table 1): (1) a natural, shallow (3 m), non-upwelling reef ("Hejie" [HJ]) located off the west coast of the Hengchun Peninsula (Figure 1C); (2) a natural, shallow, upwelling

reef ("Houbihu" [HBH]; Figure 1D; 3 m); (3) the aforementioned, shallow, upwelling and thermal effluent-influenced reef (OL; Figure 1E; 3 m); (4) a shallow, upwelling reef in the inlet of a nuclear power plant ("Inlet-inside" [ILI]; Figure 1F; 3 m); and (5) a natural, relatively deeper (7 m) upwelling reef outside of the power plant's inlet ("Inlet-outside" [ILO]; Figure 1F). To examine changes in the benthic and coral community composition before (April 2020), during (September 2020), and after (December 2020 & September 2021) the 2020 bleaching event, $35 \times 35$ cm photo-quadrat images were taken within the same triplicate 30-m $\times$ 35-cm transects at each site (~85–90 images/transect/survey time). In April 2020, September 2020, December 2020, April 2021, and September 2021, an additional three, two, two, two, and two 30 m transect surveys were deployed at OL, respectively (i.e., six, five, five, five, & five total transects, respectively, at OL ($n$ = 26 30-m transects assessed across the 18-month monitoring period). In total ~270 images/site were taken on each survey date (~300–400 images/date for OL), resulting in 7336 observations over the monitoring period (online supplemental data file (OSDF).

*2.2. Image Analysis*

For each $35 \times 35$-cm image (Figure S2), 50 random points were overlaid with Coral-Net [28], and each of the 366,800 random points was assigned one of the following 83 benthic categories. First, scleractinian corals were identified to genus level (except for *Goniopora* & *Alveopora*, which could not be differentiated by eye or CoralNet), and 34 genera were identified. For several of the more common genera, morphology was considered; montiporids were scored as branching, encrusting, or foliose, acroporids as branching or tabulate, and pavonids as massive or encrusting ($n$ = 39 groupings upon including an additional category for "unknown hard coral species"). Unless otherwise specified, *Tubastrea* spp. (sun coral), *Millepora* spp. (fire coral), and *Heliopora* spp. (blue coral) were considered in the stony coral analyses (despite not being scleractinians) since they construct reef framework in Taiwan ($n$ = 42 stony coral groups). For certain analyses, bleached corals (methods described below) in each of these groups were considered their own categories; as 13 genera were observed on the transects as bleached (+one category for "unidentifiable bleached hard corals"), there were 14 bleached groupings. Note that other genera may have bleached though were not sampled by CoralNet's artificial intelligence. In total there were 56 coral benthic groupings.

Algae were scored as one of seven bins: crustose coralline algae (CCA) on hard substrate, CCA on rubble, turf growing on hard substrate, turf growing on rubble, blue-green macroalgae, encrusting macroalgae, or upright/erect macroalgae. Seven soft coral and seven non-coral invertebrate groupings were also considered (see Appendix S2 for details), and the following six abiotic categories were quantified: dead hard coral (tissue-free skeleton that had not yet been colonized by turf algae), fine sediments (e.g., mud & silt), sand, hard substrate, transect hardware (e.g., rebar), and unclassified/unknown. Point data were first converted to feature areas within each ~0.1 m$^2$ quadrat (e.g., 20% algal cover) and then to percentages of the total ~10 m$^2$ transect area, and the transect served as the unit of replication; this involved first calculating an average across the ~90 images taken from each of the three transects (3–6 in the case of OL) at each site for each of the five survey dates (resulting in 15, 15, 26, 15, & 15 mean transect data points for HJ, HBH, OL, ILI, & ILO, respectively; $n$ = 86 pooled observations used in the statistical analyses outlined below for the 83 benthic groupings). All cover data are given as a percentage of the total benthos, though for certain analyses, they are instead presented as relative to total coral cover to emphasize differences in bleaching sensitivity among genera.

**Table 1.** The study sites, their environmental characteristics (see also Table 2), some general bleaching-related findings, and recommended conservation actions. MPA = marine protected area.

| Site Name | Hejie | Houbihu | Outlet | Inlet-Inside | Inlet-Outside |
|---|---|---|---|---|---|
| Site abbreviation | HJ | HBH | OL | ILI | ILO |
| Latitude | 21°57′22.0″ N | 21°56′17.7″ N | 21°55′54.8″ N | 21°57′16.8″ N | 21°57′11.8″ N |
| Longitude | 120°42′36.3″ E | 120°44′46.1″ E | 120°44′42.0″ E | 120°45′13.9″ E | 120°45′20.6″ E |
| Location (Figure 1A) | West Peninsula | Nanwan Bay | Nanwan Bay | Nanwan Bay | Nanwan Bay |
| Upwelling? | No | Yes | Yes | Yes | Yes |
| Thermal effluent? | No | No | Yes | No | No |
| Depth (m) | 3 | 3 | 3 | 3 | 7 |
| Fringing reef type | Natural | Natural | Artificial | Artificial | Natural |
| Most common coral genera in April 2020 (pre-bleaching; % of total benthos) | *Montipora* (14%) *Pocillopora* (12%) *Favites* (4%) | *Seriatopora* (14%) *Millepora* (10%) *Montipora* (2%) | *Montipora* (24%) *Favites* (7%) *Millepora* (5%) | *Acropora* (38%) *Pocillopora* (3%) *Montipora* (2%) | *Montipora* (11%) *Acropora* (6%) *Favites* (3%) |
| Top three thermally *susceptible* coral genera (% tissue area bleached in September 2020) | *Stylophora* (100%) *Lobophyllia* (100%) *Acropora* (98%) | *Stylophora* (100%) *Seriatopora* (91%) *Acropora* (91%) | *Stylophora* (100%) *Acropora* (89%) *Merulina* (72%) | *Merulina* (100%) *Stylophora* (100%) *Heliopora* (100%) | *Montipora* (24%) *Favites* (10%) *Porites* (7%) |
| Top three thermally susceptible coral genera (% decrease in cover: April 2020 to September 2021) Figure S3 | *Goniastrea* (100%) *Phymastrea* (100%) *Stylophora* (94%) | *Astreopora* (100%) *Goniastrea* (100%) *Merulina* (100%) | *Stylophora* (100%) *Psammocora* (100%) *Seriatopora* (100%) [a] | *Acanthastrea* (100%) *Heliopora* (100%) *Millepora* (100%) | *Fungia* (100%) *Leptoseris* (100%) *Pavona* (100%) [b] |
| Emerged on reef post-bleaching (exhaustive list) Figure S3 [c] | *Diploastrea Pachyseris Psammocora Turbinaria* | *Echinophyllia Turbinaria* | *Mycedium* | *Psammocora Tubastrea* | *Diploastrea Euphyllia Psammocora* |
| Conservation action(s) proposed herein | Establish MPA | Establish MPA & control pollution from nearby harbor | Restrict number of visitors to limit physical damage | Establish MPA | Establish MPA |

[a] Three other genera went locally extinct: *Pavona*, *Astreopora*, and *Acanthastrea*. [b] Two other genera were not observed in September 2021: *Tubastrea* and *Turbinaria*. [c] See caveat mentioned in Discussion.

**Table 2.** Seawater temperature (temp.) and community dynamics at the study sites. Note that two non-scleractinian genera, *Millepora* and *Heliopora*, were included in these calculations. Also note that *in situ* temp. loggers were deployed at different times, meaning that the "time window analyzed" in the calculation of the mean monthly maximum (MMM) differed across sites. Downwards red and upwards green arrows emphasize statistically significant (*p* < 0.02) negative and positive effects of the 2020 bleaching event, respectively. DHD = degree-heating days. DHW = degree-heating weeks. NA = not applicable. NS = not statistically significant.

| Site Name | HJ | HBH | OL | ILI | ILO | Site Effect | Temp. Effect |
|---|---|---|---|---|---|---|---|
| MMM-temp. (°C) | 28.6 | 28.3 | 29.3 | 28.0 | 28.4 | *p* < 0.0001 | NA |
| MMM-time window analyzed | 2010–2012 | 2010–2012 | 2010–2012 | 2010–2012 | 2013–2014 | NA | NA |
| Warmest month | Sept | Sept | Aug | Sept | Jul | NA | NA |
| 2020 DHWs (NOAA's Coral Reef Watch; Figure 2) | 16 | 16 | 16 | 16 | 16 | NS | NA |
| 2020 DHWs (*in situ* measurements) [a] | 16.3 | 17.0 | 16.9 | 19.9 | 1.9 | *p* < 0.0001 | NA |
| 2020 DHDs (*in situ* measurements) [b] | 101 | 104 | 108 | 120 | 13 | *p* < 0.0001 | NA |
| Mean annual temp. (°C) | 27.0 | 26.5 | 27.2 | 26.5 | 25.9 | *p* < 0.0001 | NA |
| Mean monthly temp. range (max. minus min. °C) | 3.8 | 5.5 | 7.2 | 6.1 | 6.8 | *p* < 0.0001 | NA |
| Mean temp. in warmest month of 2020 (°C) | 30.3 | 29.9 | 31.1 | 29.8 | 28.7 | *p* < 0.0001 | NA |
| Max. hourly mean seawater temp. in 2020 (°C) | 31.8 | 32.2 | 35.6 | 32.7 | 31.3 | *p* < 0.0001 | NA |
| Summed time above 30 °C in 2020 (days) | 47 | 33 | 78 | 25 | 7 | *p* < 0.0001 | NA |
| # days in 2020 with mean temp. above 30 °C | 42 | 32 | 72 | 22 | 2 | *p* < 0.0001 | NA |
| Heat accrual interval (2020) | 3 Jun–1 Oct | 11 Jun–1 Oct | 27 Mar–14 Oct | 13 Jun–3 Oct | 12 Jun–16 Sept | NA | NA |
| Coral cover before bleaching event (%) [b] | 47 | 37 | 55 | 44 | 36 | NS | NA |
| Coral cover after bleaching event (%)-Dec 2020 [b] | 22▼ | 7.5▼ | 22▼ | 11▼ | 48 | *p* < 0.0001 | *p* < 0.0001 |
| Coral cover after bleaching event (%)-Sept 2021 [b] | 28▼ | 18▼ | 51 | 46 | 52 | *p* < 0.01 | *p* < 0.001 |
| % hard coral cover increase/decrease (18 months) | −36%▼ | −49%▼ | −7% | +7% | +44% | *p* < 0.0001 | *p* < 0.0001 |
| % of all corals that were bleached in Sept 2020 (Figure 4) [c] | 74 | 83 | 65 | 97 | 8.5 | *p* < 0.001 | *p* < 0.0001 |
| Mean bleaching percentage across genera [d] | 78 | 84 | 57 | 64 | 3.7 | *p* < 0.001 | *p* < 0.0001 |
| # genera present: Apr 2020➜Sept 2021 | 220➜24 | 210➜20 | 220➜17 | 170➜16 | 280➜26 | *p* < 0.001 | NS |
| Shannon diversity: April 2020➜Sept 2021 | 2.20➜2.0 | 1.70➜1.8 | 1.90➜1.1▼ | 0.60➜0.3 | 2.30➜2.2 | *p* < 0.001 | NS |
| Evenness: April 2020➜Sept 2021 | 0.70➜0.7 | 0.60➜0.6 | 0.70➜0.4▼ | 0.20➜0.1 | 0.80➜0.7 | *p* < 0.001 | NS |
| Coral/algae ratio: April 2020➜Sept 2021 | 1.40➜0.6▼ | 1.50➜0.7▼ | 2.90➜3.1▲ | 4.00➜2.6▼ | 1.60➜T3.1▲ | *p* < 0.0001 | NS |

[a] See Figure 3A–E. [b] See Figure 3F–J. [c] Note the distinction between these values and those of Figure 3, which instead present the percentage of bleached corals over the entire benthos. [d] Note the distinction between these values and those of Figure 4, in which the transect (rather than the genus) was the replicate.

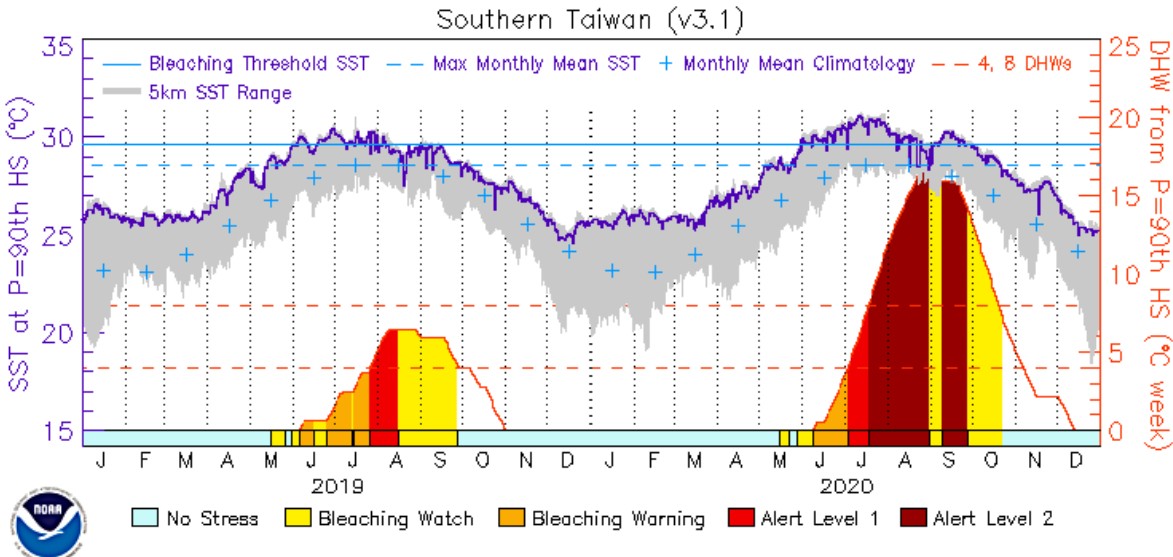

**Figure 2.** Sea surface temperature (SST; °C; left *y*-axis), degree-heating weeks (DHWs; right *y*-axis), and coral bleaching alert benchmarks in Southern Taiwan from January 2019 to December 2020, as calculated by the United States' National Oceanic and Atmospheric Administration's Coral Reef Watch (CRW; https://coralreefwatch.noaa.gov/; accessed on 31 January 2021). The horizontal orange lines at four and eight DHWs reflect the window within which bleaching typically begins to occur based on CRW's historical predictions.

Given the random nature of CoralNet's image sampling algorithm, bleaching was not assessed by tracking the fates of individual coral colonies; instead, the percentages of coral tissue areas sampled by CoralNet that had bleached (confirmed by eye following a previous method [29]) were calculated across each transect as a whole, as well as at the genus level. Bleaching was reported two ways: (1) the mean total percentage of all corals bleached per transect or (2) the mean % bleached averaged for each bleaching-sensitive genus; note that, in some cases (Table 2), these values differ somewhat since the former is relative to the entire benthos, whereas the latter is based on a genus-by-genus assessment. Similarly, mortality could not be calculated for each colony since different colonies may have been sampled by CoralNet at each survey time within each photo-quadrat; instead, it was inferred from relative decreases in cover of the reef inhabitants between April 2020 and September 2021. Live coral cover was presented as either unbleached coral cover only or as bleached + unbleached (but excluding colonies that had recently perished). Because some coral genera, in particular, *did* go locally extinct over the duration of the study, we calculated both Shannon diversity and species evenness and tested whether either (in addition to simple alpha [generic] diversity) changed over time.

*2.3. Statistical Analyses*

All statistical analyses were undertaken with the JMP® Pro 16–17 suite (Cary, NC, USA). We used two-way ANOVA to assess the effects of survey time (month; df = 4), site (df = 4), time × site, and transect(site × month) on coral cover and bleaching percentage (Table S1); the latter was treated as a random effect in a mixed model. Note that since the 2020 bleaching event was so much more severe than the 2021 one, we typically parsed these univariate ANOVAs by survey year except for when directly testing recovery-associated processes. For certain analyses, we considered depth as a factor: four shallow sites (3 m) vs. one deeper site (7 m). We also tested for the effects of upwelling (df = 1: upwelling [*n* = 4] vs. non-upwelling [*n* = 1]) and reef type (df = 1): natural (*n* = 3) vs. artificial (*n* = 2; ILI & OL; see Table 2.).

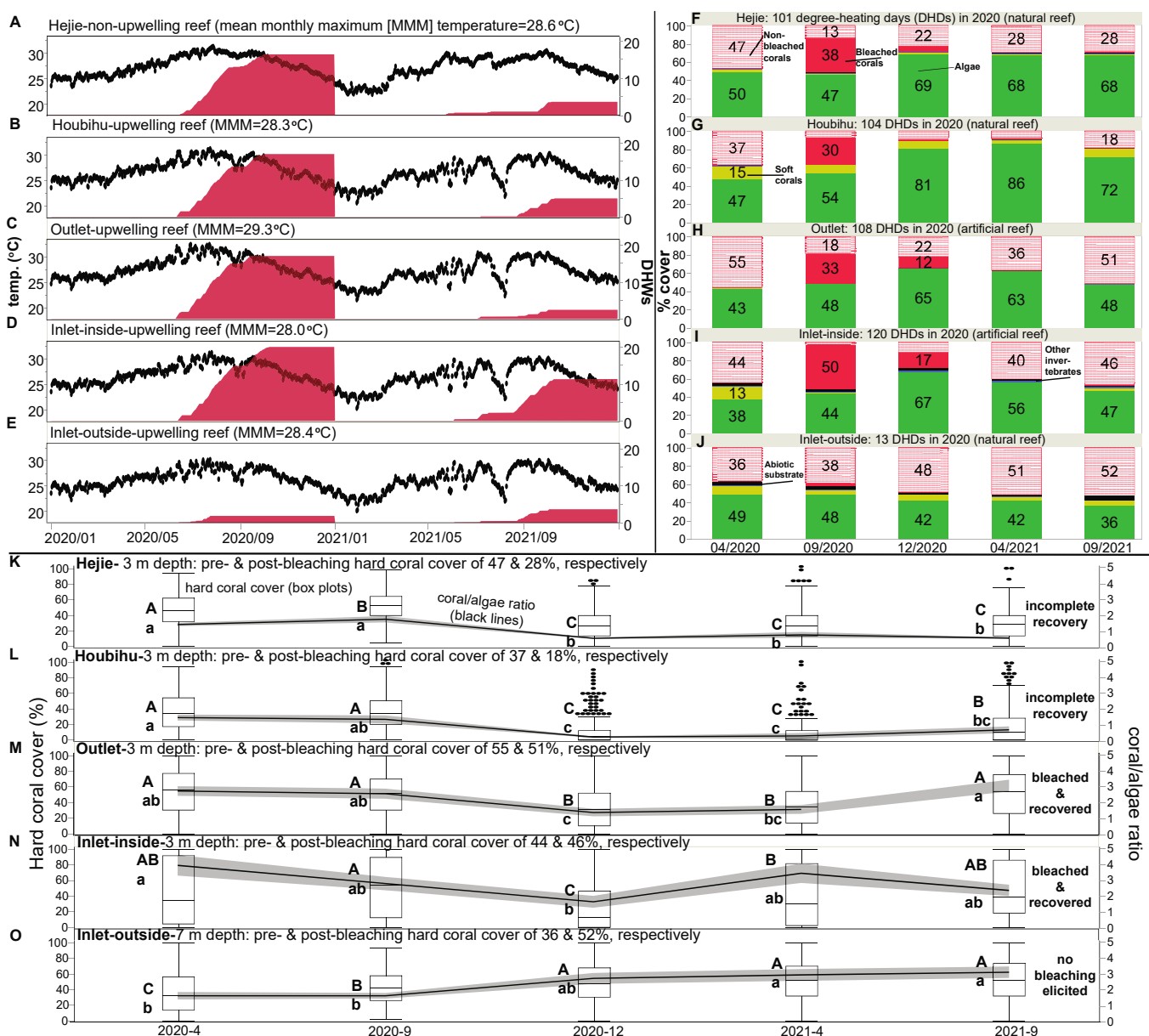

**Figure 3.** Annual temperature profiles of the fives study sites *in situ* and overall benthic composition. Both raw temperature (left y-axes of (**A–E**)) and degree-heating weeks (DHWs; right y-axes of (**A–E**)) have been shown. In the stacked bar charts (**F–J**), the two non-scleractinian reef-builders (*Millepora* spp. & *Heliopora* sp.) have been grouped with the hard corals: "non-bleached corals" (light red with hatches) and/or "bleached corals" (solid red). Hard coral cover (left y-axes of (**K–O**)) and the coral/algae ratio (right y-axes of (**K–O**)) are shown as box plots and black lines, respectively; the shading about the latter corresponds to SEM, and the upper- and lowercase letters correspond to Tukey's *post-hoc* differences for hard coral cover and the coral/algae ratio, respectively (*p* < 0.02). With the exception of Inlet-outside (ILO; (**J,N**)), all benthic communities changed in structure over time (Table 3).

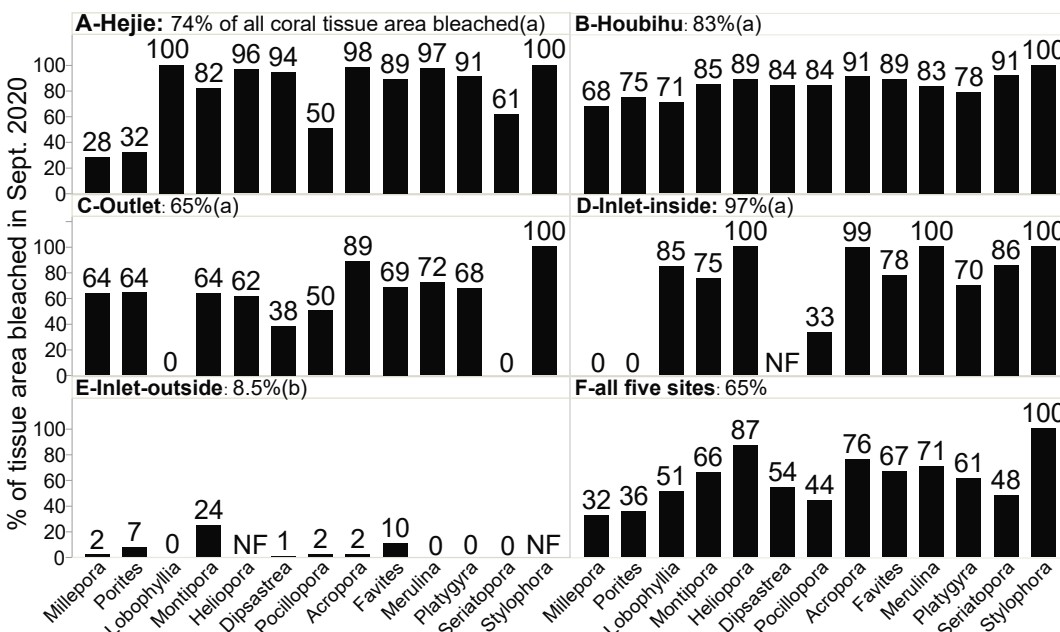

**Figure 4.** Bleaching patterns across the 13 most common coral genera found at the five study sites. Note that these values represent the mean tissue area bleached in September 2020 (**A–E**), rather than the percentage of surveyed colonies that presented bleached tissues. As there was a significant effect of site on overall bleaching percentage (one-way ANOVA, $p < 0.0001$), the lowercase letters following the site means adjacent to the panel labels reflect Tukey's *post-hoc* differences ($p < 0.05$). Averages for each genus across the five sites have been shown in (**F**). NF = genus not found at site.

For the coral community composition (to genus level), a similarity matrix was constructed using Euclidean distances derived from raw percent abundance values. After determining the minimum number of multi-dimensional scaling (MDS) dimensions with a stress <0.1, a non-parametric multivariate ANOVA (NP-MANOVA) was undertaken with the 2–6 dimensions (stress = 0.04–0.1) as Y and the same model as described for coral cover as the predictors. JMP's "predictor screen," which is based on random tree bootstrap forests ($n = 100$ trees), was used to identify the univariate trend that contributed the most variation to the multivariate effects documented by NP-MANOVA. This analysis was repeated for each site in isolation, albeit with only month as a fixed effect. In these analyses, we also tested for the effects of transect and transect(month).

We also considered the effects of survey month, year, site, upwelling, and reef type on overall percentages of bleached tissues to determine which most strongly affected the bleaching response via univariate NP-ANOVAs, student's *t*-tests, or Wilcoxon tests (depending on the number of factors & the distribution + variance of the data); in general, this analysis was carried out separately for each of the two survey years since the degree of bleaching in 2020 was far greater than in 2021 (when few DHWs were documented; Figure S1). To construct a more comprehensive, holistic model of the environmental drivers of the coral community assemblage changes documented by NP-MANOVA, we used partial least squares (PLS) with the environmental predictors as X's and the 83 benthic categories as Y's. We used the JMP Pro 16-recommended VIP cutoff of 0.8 for influential model term inclusion, and overlaid the 25 month ($n = 5$ survey months) × site ($n = 5$ study sites) interaction groups (i.e., pooled across replicate transects for each site) onto a two-dimensional correlation loading plot to approximate a distance-based redundancy analysis (dbRDA) that permitted visualizing relationships among all environmental predictors and benthic taxa/categories assessed (i.e., distance-based linear modeling). The secondary goal of this analysis was to determine the degree to which the measured predictors (e.g., artificial vs. natural reefs) contributed to variation in the benthic assemblage over space and time.

**Table 3.** Non-parametric multivariate ANOVA (NP-MANOVA) of benthic community structure. In this analysis, 56 reef-building coral genera (including *Tubastrea*, *Heliopora*, & *Millepora*), seven algal groupings, seven soft coral groupings, seven invertebrate (non-coral) groupings, and six abiotic groupings were considered (*n* = 83 categorical groupings) in a multi-dimensional scaling (MDS) analysis (Euclidean distance-based similarity) that featured data from all sites ("All five sites"), and the first four sets of coordinates were used in an NP-MANOVA. Separate, site-by-site NP-MANOVAs were conducted in a similar manner, albeit with fewer total benthic groupings since not all taxa were found at all sites. "Type" refers to natural (*n* = 3) vs. artificial reefs (*n* = 2). Statistically significant differences (alpha = 0.02) have been highlighted in bold, and the "greatest contributor to variation" associated with the respective row effect was derived from a bootstrap forest-based predictor screen with 100 trees. CCA = crustose coralline algae.

| Effect | Wilks' Lambda | Approx./ Exact F | p | Greatest Contributor to Variation (% Variation Explained) |
|---|---|---|---|---|
| **All five sites**-83 benthic groupings, 4 MDS dimensions, and MDS stress of 0.08 (*n* = 86 data points) | | | | |
| site | 0.02 | 26.87 | **<0.0001** | Foliose *Montipora* spp. cover significantly higher at OL (27%) |
| month | 0.08 | 13.56 | **<0.0001** | Encrusting macroalgae significantly more common in Apr. (both years; 27%) |
| site × month | 0.05 | 4.21 | **<0.0001** | Foliose *Montipora* spp. cover significantly higher at OL in all months (21%) |
| depth | 0.21 | 3.70 | **<0.01** | *Sarcophyton* sp. abundance was significantly higher at 7 m (43%) |
| depth × month | 0.66 | 2.01 | **0.01** | CCA on hard substrate significantly more abundant at 3 m in 2021 (22%) |
| type | 0.31 | 5.65 | **<0.001** | Significantly higher abundance of *Lobophyton* sp. on natural reefs (16%) |
| type × month | 0.67 | 1.90 | 0.02 | |
| upwelling | 0.42 | 4.50 | **<0.0001** | *Pocillopora* spp. significantly more abundant on the non-upwelling reef (69%) |
| upwelling × month | 0.93 | 0.34 | 0.99 | |
| transect (site × month) | 0.41 | 0.09 | 1.00 | |
| **Hejie (HJ; shallow, natural, non-upwelling reef)**-71 benthic groupings, 2 MDS dimensions, and MDS stress of 0.06 (*n* = 15) | | | | |
| month | 0.002 | 45.29 | **<0.0001** | Significantly more bleached soft corals in December 2020 (11%) |
| transect | 0.98 | 0.04 | 1.00 | |
| transect (month) | 0.89 | 0.02 | 1.00 | |
| **Houbihu (HBH; shallow, natural, upwelling reef)**-66 benthic groupings, 3 MDS dimensions, and MDS stress of 0.05 (*n* = 15) | | | | |
| month | 0.002 | 16.15 | **<0.0001** | Significantly more foliose *Montipora* spp. in April and September 2020 (13%) |
| transect | 0.87 | 0.24 | 0.96 | |
| transect (month) | 0.48 | 0.06 | 1.00 | |
| **Outlet (OL; shallow, artificial, effluent-infused, upwelling reef)**-61 benthic groupings, 3 MDS dimensions, and MDS stress of 0.09 (*n* = 25) | | | | |
| month | 0.05 | 7.97 | **<0.0001** | Bleached *Montipora* spp. significantly more common in Sept. 2020 (14%) |
| transect | 0.70 | 0.43 | 0.96 | |
| transect (month) | 0.10 | 0.13 | 1.00 | |

**Table 3.** *Cont.*

| Effect | Wilks' Lambda | Approx./ Exact F | p | Greatest Contributor to Variation (% Variation Explained) |
|---|---|---|---|---|
| **Inlet-inside (ILI; shallow, artificial, cooler upwelling reef)**-61 benthic groupings, 2 dimensions, and MDS stress of 0.09 ($n = 15$) | | | | |
| month | 0.06 | 6.74 | **<0.001** | More hard substrate (abiotic) in April. (both years; 16%) |
| transect | 0.66 | 1.29 | 0.30 | |
| transect (month) | 0.43 | 0.16 | 1.00 | |
| **Inlet-outside (ILO; deep, natural, upwelling reef)**-69 benthic groupings, 4 MDS dimensions, and MDS stress of 0.07 ($n = 15$) | | | | |
| month | 0.001 | 7.11 | **<0.001** | More CCA on hard substrate in 2021 vs. 2020 (14%) |
| transect | 0.05 | 4.32 | 0.02 | |
| transect (month) | 0.08 | 0.14 | 1.00 | |

*2.4. Seawater Temperature*

Temperature was recorded every 10 min at each site at 3 m (all sites except ILO [7 m]) with HOBO® Pendent loggers (USA). Temperature was assessed over space (site) and time (month) via two-way ANOVA (alpha = 0.02 for all analyses). Both degree-heating days (DHDs) and DHWs were calculated based on the mean monthly maximum (MMM) over 90-day periods to quantify heat accumulation (*sensu* [30,31]). Although NOAA's Coral Reef Watch (CRW) currently uses a trailing MMM of approximately 28.6 °C based on sea surface temperatures (SST), we instead re-calculated site-specific MMM using our *in situ* temperature data (Figure S1); note that the MMM differed significantly across sites given their markedly different oceanographic conditions (Table 1). Although the MMM + 1 °C bleaching threshold would range from 29.0–30.3 °C based on our *in situ* values, experimental works have shown this threshold to be higher for certain species from local upwelling reefs (closer to 31–32 °C [32,33]). We also extracted DHW data over the 2020 bleaching event (Figure 2) from NOAA's CRW website (https://coralreefwatch.noaa.gov/; accessed on 31 January 2021). It is worth mentioning that the oceanography of Nanwan Bay has been extensively characterized, particularly with respect to large-amplitude internal waves [20,27] and their effects on the physiological responses of the local corals [34,35].

**3. Results**

*3.1. Temperature*

Based on a trailing MMM of ~29 °C, NOAA's satellite data-based estimates calculated 16 DHWs for Southern Taiwan's reefs in 2020 (Figure 2 & Table 2). This value agrees well with DHW calculations made from our at-depth *in situ* temperature data (Figure 3A–E and Figure S1) for the four shallow reefs (DHWs ranging from 16–20); however, the seawater was significantly cooler at 7 m (ILO), and this site experienced only 2 DHWs in 2020 (Table 2). Although ILO was not characterized by the lowest MMM (Table 2), it was the coolest site overall (annual mean = 25.9 °C), and the temperature never surpassed the empirically derived local coral bleaching threshold of 31.5 °C determined previously [32,33]. The MMM of the OL, 29.3 °C, was significantly higher than that of the other reefs; as such, although the DHWs of the four shallow, fringing reefs were similar (Table 2), the actual degree of thermal dosing at OL was far higher due to the regular inundation with warm effluent water from the nearby nuclear power plant [36,37]. Corals of the OL were exposed to temperatures >30 °C for ~78 days across a ~six-month period in 2020, and the temperature even reached 35 °C for several hours (Table 2); there were 72 days in 2020 in which the mean temperature was above 30 °C. In fact, all shallow reefs were immersed in seawater >30 °C for at least 25 days over the June–October 2020 period; this degree of thermal dosing was clearly sufficient to induce mass coral bleaching at all but the deeper reef.

*3.2. Benthic Community Changes in Response to the 2020 Marine Heatwave*

Prior to the 2020 bleaching event (April 2020), the coral/algal cover of the four shallow reefs—HJ (Figure 3F), HBH (Figure 3G), OL (Figure 3H), and ILI (Figure 3I)—averaged 47/50% (0.9:1; Figure 3K), 37/47% (0.8:1; Figure 3L), 55/43% (1.3:1; Figure 3M), and 43/38% (1.2:1; Figure 3N), respectively (mean coral & algal covers of 3-m reefs of 47 & 44%, respectively; 1.1:1). By the time of the during-bleaching survey (September), >65% of these corals had bleached (Figure 4A–D), and the algal cover had risen to 54, 48, and 44% at HBH (Figure 3G), OL (Figure 3H), and ILI (Figure 3I), respectively; algal cover remained similar at HJ (~47%; Figure 3F). Univariate trends in overall hard coral cover and bleaching over space and time can be found in Table S1; the site x time interaction effects are instead discussed herein.

Upon resurveying these reefs in December 2020, 53, 80, 60, and 74% decreases in coral cover were documented at HJ, HBH, OL, and ILI, respectively, and the coral covers of these reefs were only 22 (Figure 3F), 7.5 (Figure 3G), 22 (Figure 3H), and 11% (Figure 3I), respectively. Note that at this time, there were still live bleached corals on these four reefs (6, 3, 12, & 17% of total benthic cover, respectively). Over the course of 2021, coral cover

remained 36 and 49% lower than in April 2020 at HJ (30% cover in December 2021, of which 2% were still bleached) and HBH (19% in December 2021, of which 1% were still bleached), respectively. In contrast, coral cover nearly reached the pre-bleaching level of 55% at OL (51% in September 2021), and coral cover was actually higher at ILI: 47% in Sept. 2021 (46% in Figure 3I since 1% of these 47% were still bleached.) vs. 44% in Apr. 2020. The 7-m site, ILO, showed a different trend (Figure 3J): healthy coral and algal covers were 36 and 49%, respectively, in April 2020 (0.7:1; Figure 3O), yet only 9% of the coral tissue area was bleached in September 2020 (Figure 4E). Hard coral cover (unbleached corals) actually increased to 38% in September 2020, with algal cover staying relatively consistent (~50%). By the time the heat wave had passed (December 2020), coral cover had risen to nearly 50% at ILO, reaching ~52% by the final survey time (September 2021).

### 3.3. Changes in Dominant Reef Coral Genera

When considering the benthic data at a finer level of taxonomic resolution (corals to genus level), there were clear shifts in the coral community over time for all sites (Figures 4–7 & Tables 1–3). We now discuss these in series for each of the study sites.

### 3.3.1. Hejie (HJ)

At the non-upwelling site, the entire benthic community changed significantly over time (Table 3), and *Montipora* (14%) and *Pocillopora* (11.5%) were the dominant coral genera pre-bleaching (Table 1); for these data, as well as those of the remaining four sites, please see Figure 5 for values expressed as relative to the top five most common coral genera and Figure 6 for raw values of the benthic cover for the top three most common coral genera at each site. During the Sept. 2020 bleaching event, 100% of surveyed *Lobophyllia* and *Stylophora* tissue areas were bleached (Figure 4A), the highest of all genera. Montiporids also underwent significant bleaching (82%; Figure 4A), which resulted in a statistically significant decline in cover over time: from 14% pre-bleaching to <5% cover by the final survey time (Figure 6A). In contrast, only 50% of pocilloporid tissue area was bleached, and they had supplanted *Montipora* as the most dominant coral genus by December 2020 (12% of total benthic cover [Figure 6A], constituting half of all coral cover, or 65% of the top five genera [Figure 5A]). Even fewer poritids bleached (32% of tissue area surveyed; Figure 4A), and the poritid cover was the same pre- and post-bleaching (2–3% of the benthos; OSDF). Although the number of coral genera observed increased slightly over time (from 22 genera in April 2020 to 24 September 2021; Table 2), both Shannon diversity and evenness decreased over this period (Figure 7A); note that neither decrease was statistically significant.

### 3.3.2. Houbihu (HBH)

At the shallow, natural, upwelling reef HBH, the dominant coral genera prior to bleaching (Table 1) were *Seriatopora* (14% of the benthos [Figure 6B], nearly 70% of all hard corals, or 69% of the top five coral genera [Figure 5B] and *Millepora* (10% of the benthos [Figure 6B]), and the benthic community structure changed significantly over time (Table 3). The most bleaching-susceptible genera (Figure 4B) were *Stylophora* (100% of CoralNet-sampled tissue areas bleached), *Seriatopora* (91% bleached), and *Acropora* (91%). Additionally, 100% decreases in cover were documented for the following genera: *Astreopora*, *Goniastrea*, and *Merulina* (Figure S3); it is likely that this was due to bleaching-associated mortality, though note that individual columns were not fate-tracked per se. There was also a 100% decrease in cover of foliose montiporids (OSDF).

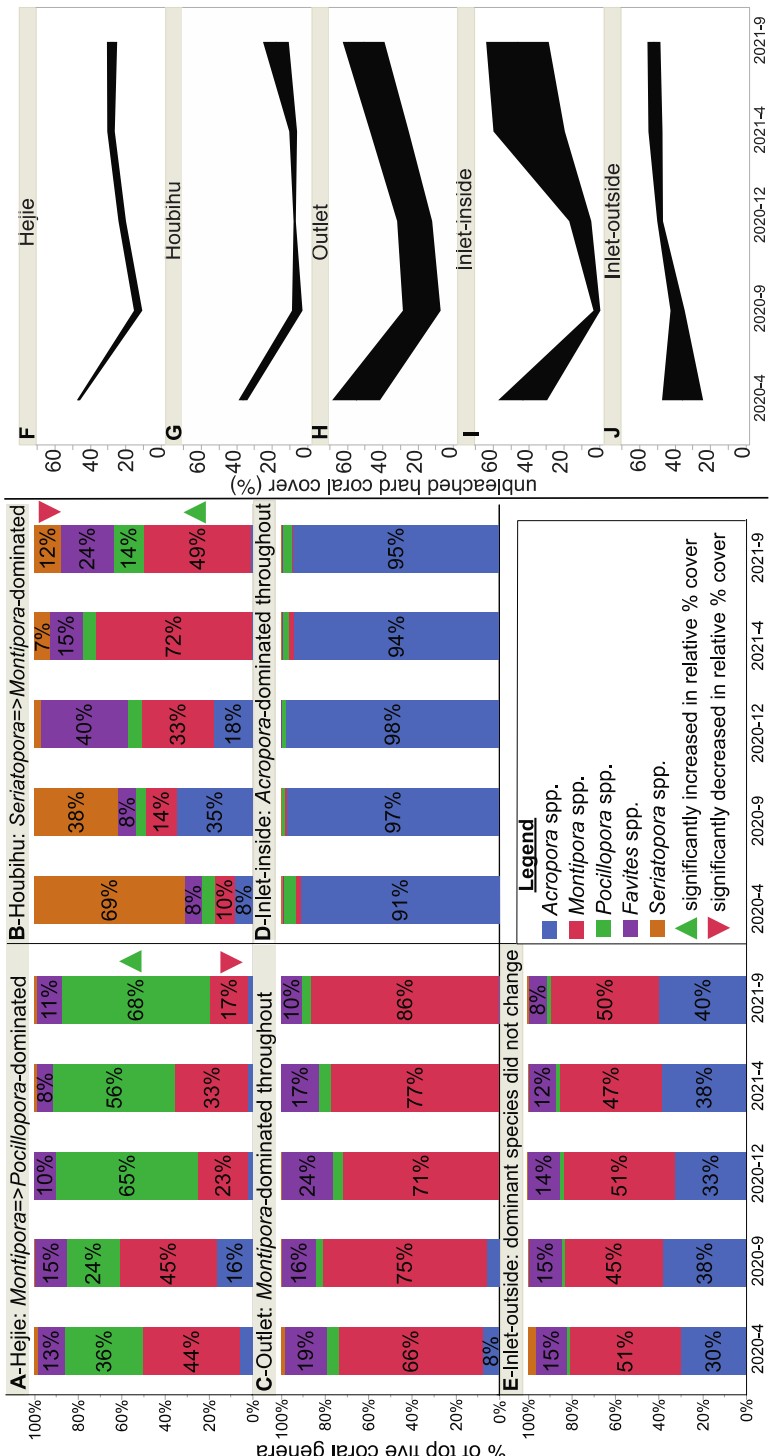

**Figure 5.** Relative percentages of the top-five coral genera of the five study sites (**A–E**). Note that the values are presented as relative to the total cover of these five genera, rather than as a percentage of the total benthos. The prevailing trend over the 18-month study period has been stated adjacent to the site name. Upwards-facing green and downwards-facing red arrows denote statistically significant increases and decreases, respectively, in relative proportions over time. Panels (**F–J**) depict from unbleached coral cover percentage data from Figure 3 over time for comparative purposes, and shading depicts standard deviation about an obscured mean line. See Figure 6 for raw percent changes relative to the entire benthos.

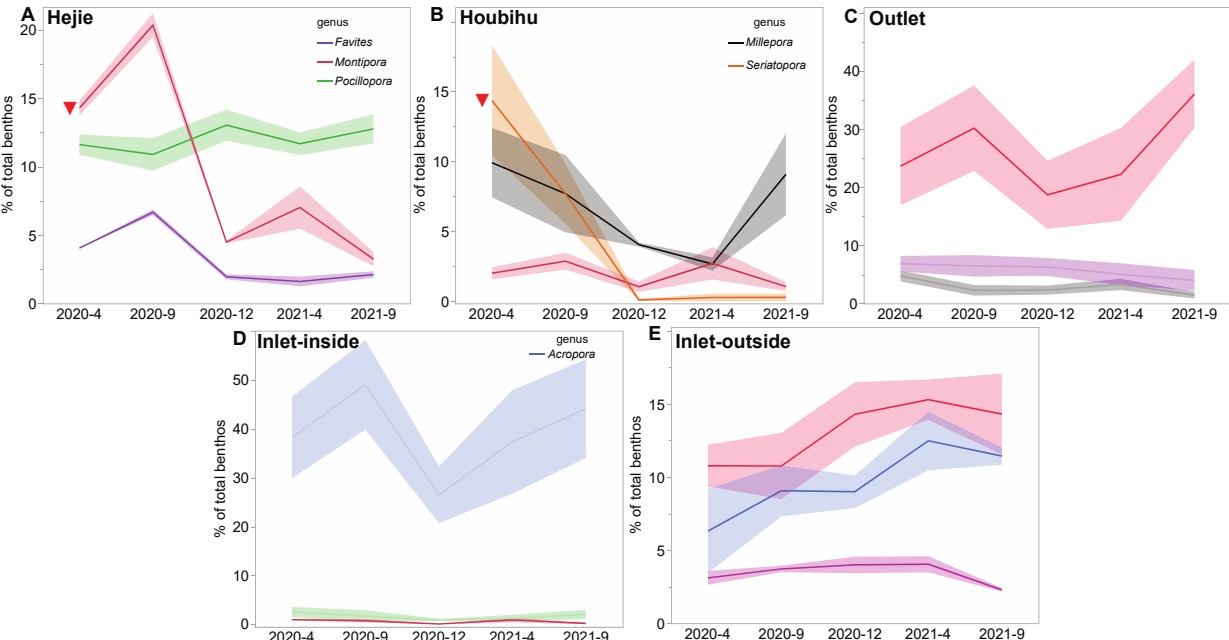

**Figure 6.** Change in cover of the top three coral genera of each site (**A–E**) over the 18-month survey period. Note that bleached corals that had not yet died were included in this analysis. The color corresponding to the respective genus is shown only in the first panel in which the associated data are found and extends to all further panels featuring that genus. Values represent site means at each survey time, with the shading encompassing standard error of the mean. When the abundance of a genus decreased significantly over time (one-way ANOVA, $p < 0.02$), a downwards-facing arrow has been presented next to the April 2020 data.

As *Seriatopora* cover dropped dramatically over time as a result of bleaching (Figure 6B), *Millepora* became the most common coral after bleaching; however, its cover nevertheless dropped from 10 to 9% over the monitoring period (Figure 6B). *Heliopora* was among the most thermotolerant genera; cover increased from 1.3 to 1.8% despite 90% of tissue area having been bleached in September 2020 (OSDF). Generic diversity (Table 2), Shannon diversity, and evenness (Figure 7B) all decreased over time as a result of bleaching, though these differences were not statistically significant. Hard coral cover had not returned to pre-bleaching levels (37%) by September 2021 (<18%; Figure 3G).

### 3.3.3. Outlet (OL)

At the OL, *Montipora* (foliose morphology) was the most common genus prior to bleaching (Table 1): 17.5% of the benthos (Figure 6C) or 66% of the top five coral genera (Figure 5C). *Stylophora* and *Acropora* demonstrated the highest percentage of CoralNet-sampled tissue areas bleached (100 & 89%, respectively; Figure 4C). However, they were not the genera whose cover declined the most over time; the following genera actually went locally extinct (Figure S3): *Acanthastrea*, *Astreopora*, *Pavona*, *Psammocora*, *Seriatopora*, and *Stylophora*. Their absence in December 2021 led to a statistically significant change in the benthic community structure over time (Table 3). Although 64% of tissue areas were bleached in September 2020, and foliose morphology cover dropped by 59%, *Montipora* remained the most abundant genus by September 2021 (~35% of the benthos [Figure 6C] or 86% of the top five coral genera [Figure 5C]). This is because encrusting and branching morphologies actually increased in abundance (from 5.5 to 6.5% & from 0 to 1.5%, respectively). No tissues areas of *Lobophyllia* spp. were bleached in September 2020, yet cover nevertheless decreased by 31% over the 18-month period (Figure S3). Given the local extinction mentioned above, as well as this dramatic decline in *Lobophyllia* abundance, generic diversity (Table 2), Shannon diversity, and evenness (Figure 7C) all decreased over

time, and the latter two decreases were statistically significant ($p < 0.001$); the evenness of the top five coral genera, however (Figure 5C), stayed relatively consistent over time.

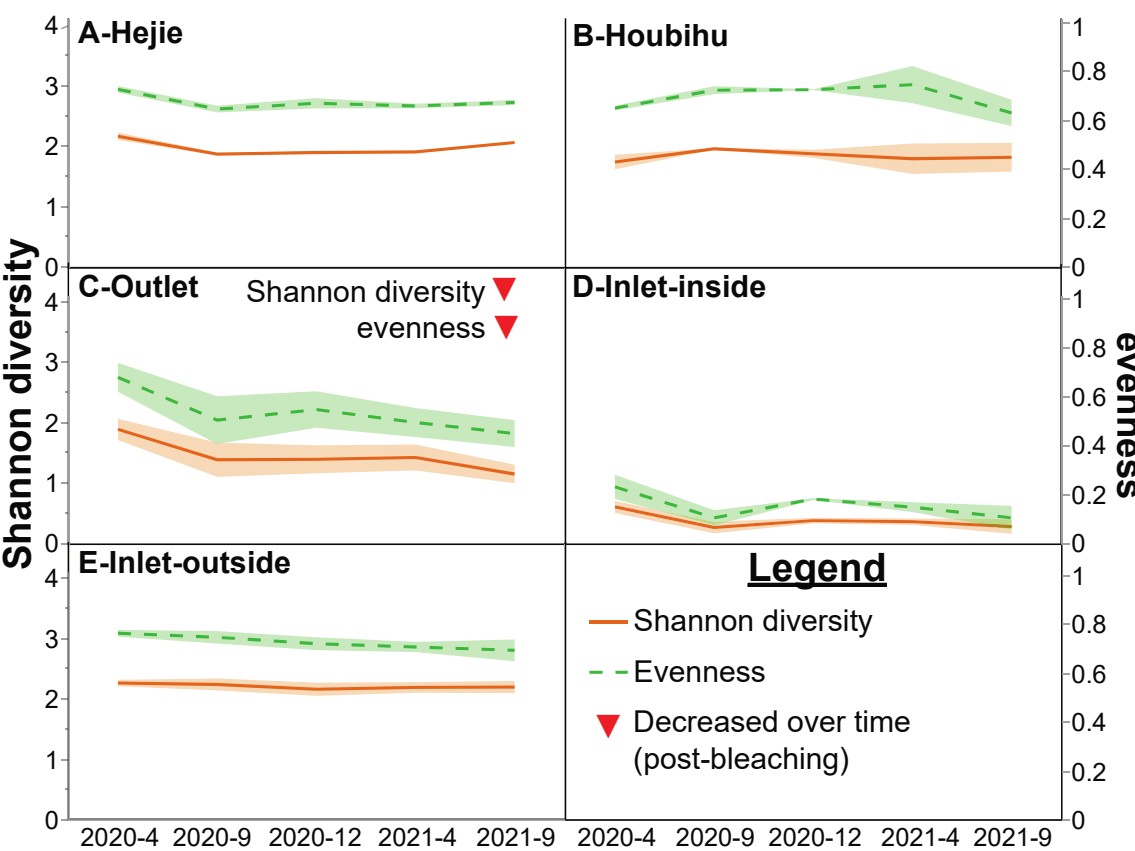

**Figure 7.** Shannon diversity and coral species evenness over time for each of the five study reefs (**A–E**). Scleractinian corals, *Heliopora* spp., and *Millepora* spp. were considered in the calculations, and error bands reflect SEM. When either decreased significantly over time ($p < 0.02$), a downwards-facing red arrow has been placed next to the respective parameter. Please see Figure S3 for genus-by-genus relative abundance changes over time at the five field sites.

### 3.3.4. Inlet-Inside (ILI)

At ILI, *Acropora* (37% of benthic cover [Figure 6D]) was the most abundant genus in April 2020 (91% of the cover of the top five coral genera [Figure 5D]). However, it demonstrated amongst the highest incidences of bleaching; 99% of tissues imaged and sampled by CoralNet were bleached in September 2020 (Figure 4D). No tabular acroporids remained in December 2020, and the branching acroporids had underwent a 75% decrease in cover (Figure S3 & OSDF). Nevertheless, acroporids still comprised ~25% of the benthic cover in December 2020 (Figure 5D), though ~65% of the surviving acroporids were still in a bleached state at that time (OSDF). Acroporids compromised >90% of the top five coral genera at all times (Figure 5D), and, by the final survey time, *Acropora* cover had actually surpassed pre-bleaching levels (>40%; Figure 6D; note that this increase was not statistically significant.). No poritid corals bleached, and their cover actually increased (albeit from only 0.05 to 0.1%) over the remainder of 2020 before declining over 2021 back to 0.02% (Figure S3 & OSDF). There was a statistically significant multivariate change in the benthic community structure (Table 3), and generic diversity (Table 2) decreased post-bleaching; neither Shannon diversity (Figure 7D) nor evenness changed significantly over time (Figure 7D) at this overall lower diversity site (only 16–17 coral genera present).

### 3.3.5. Inlet-Outside (ILO)

At the 7-m site, *Montipora* (11% of the benthos [Figure 6E]) and *Acropora* (6%) were the dominant genera prior to bleaching (Table 1), collectively comprising >80% of the top five genera (Figure 5E). *Montipora* was the genus that bleached most severely, with 24% of surveyed tissues bleached in September 2020 (Figure 4E). Several genera that were observed in April 2020 were absent in the September 2021 surveys (100% drop in cover; Figure S3): *Fungia*, *Leptoseris*, *Pavona*, *Tubastrea*, and *Turbinaria*. Unlike the 3-m sites, the benthic community structure of ILO did not change significantly over time given the overall absence of bleaching (Table 3); on average < 10% of sampled coral tissue areas were bleached in September 2020, versus 65–97% for the shallow sites (Figure 4). Generic diversity, however, did decrease slightly (Table 2); in contrast, both Shannon diversity and evenness (Figure 7E) increased over time despite the aforementioned absence of five coral genera in September 2021. Neither difference was statistically significant.

### 3.4. Inter-Site Analysis

When averaging across survey months in each year (Figure 8), there was not a statistically significant effect of site (Figure 8A) on bleaching prevalence for either survey year (one-way ANOVA, $p > 0.02$). However, when looking *only* at the data from September 2020 (when the bleaching event was ongoing), there was a statistically significant site effect ($p < 0.001$) on percentage of sampled coral tissue areas bleached; see Figure 4 for the inter-site *post-hoc* comparisons. Bleaching prevalence was significantly more widespread in September 2020 (Figure 8B) when compared to other months, though 30% of corals were still bleached in December 2020. When averaging across survey times within each year, bleaching prevalence was not significantly higher in the shallow reefs versus the deep one (Figure 8C); only when analyzing the September 2020 bleaching event data alone was a depth effect evident (student's *t*-test, $p < 0.0001$; 78 & 8.5% of coral tissues sampled at 3 & 7 m, respectively, were bleached.). In contrast, the bleaching prevalence did not differ between the three natural and two artificial reefs, either when pooling across survey times within year ($p = 0.11$; Figure 8D) or when looking at the September 2020 data only (56 & 77%, respectively; $p = 0.17$). However, this lack of significance is because one of the three natural reefs, ILO, generally resisted bleached. Recovery was far more pronounced at the two artificial reefs vs. the two natural reefs located at the same depth (3 m).

Bleaching prevalence was similar in upwelling and non-upwelling reefs, both when pooled across survey times in 2020 ($p = 0.98$; Figure 8E) and when looking at the September 2020 data only (student's *t*-test, $p = 0.61$; 74 & 64% of coral tissue areas of non-upwelling & upwelling reefs, respectively, were bleached). Unlike for the other predictors (Figure 8F-I), there was a significant effect of upwelling for the pooled 2021 data (Figure 8J); bleaching prevalence was over two times higher on the non-upwelling reef (HJ; 5%) versus the four upwelling reefs (2%). It is important to note that bleaching was far less widespread in 2021. Furthermore, as only one non-upwelling reef was included in this analysis, it is premature to validate the hypothesis that, all else being equal, corals from upwelling reefs will fare better under rising temperatures [32,33]. When considering all possible drivers of bleaching susceptibility in the same model with PLS (Figure 8K), all environmental predictors considered—site, month, year, depth, upwelling, and type (natural vs. artificial)—were associated with VIP > 0.8 and consequently kept in the final, six-(latent) factor model for which the root mean PRESS was minimized (1.1) using a NIPALS algorithm (kfold validation = 7). That said, this dbRDA explained only 22% of the variation in the assemblage across the first two dimensions (Figure 8K), and even the entire, six-factor model explained only 49% of the cumulative variation in the benthic assemblage; additional environmental factors that went unmeasured must explain the remaining 51% of the variation associated with the changing coral assemblages across this bleaching event.

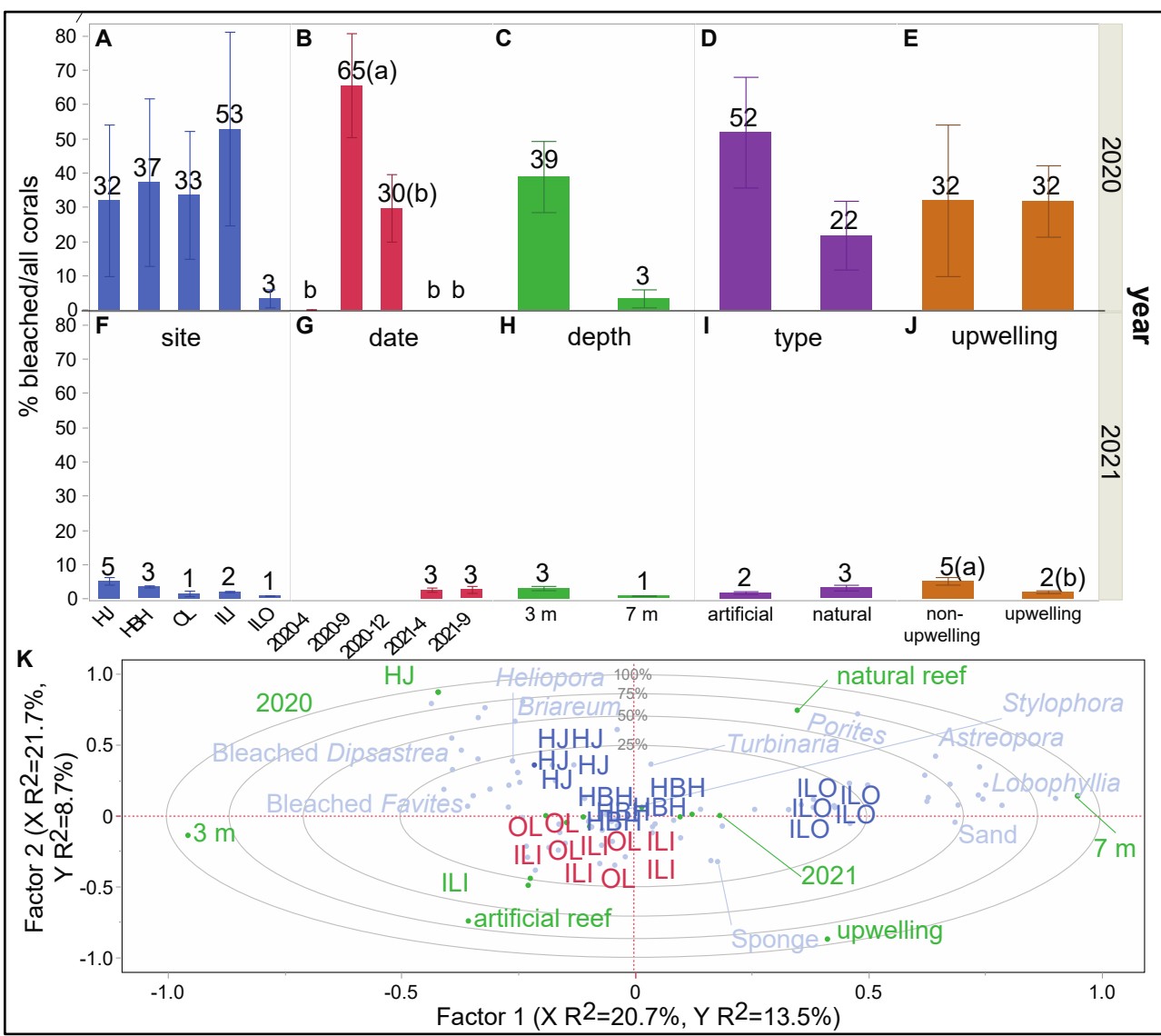

**Figure 8.** The effects of various environmental predictors on bleaching prevalence (% of all corals bleached) and a partial least squares (PLS) analysis of drivers of changes in the coral community over time. Error bars in (**A–J**) represent SEM, and lowercase letters in parentheses following percentage values in (**B,J**) reflect Tukey's honestly significant inter-mean (*post-hoc*) differences ($p < 0.05$; only carried out when a significant effect was documented in the non-parametric ANOVA or *t*-test). In the PLS correlation loading plot (**K**), only select X (environmental predictors; green text & dots) and Y (benthic categories; light blue text & dots) have been shown due to spatial constraints, and the means for each site x month interaction group ($n = 25$) have been presented (see main text for site abbreviations.); groups highlighted in dark blue and red represent natural ($n = 3$) and artificial ($n = 2$) reefs, respectively. Only the first two of six factors have been shown.

## 4. Discussion

The 2020 marine heatwave was associated with nearly 17 DHWs, far higher than the 7–11 values that elicited bleaching in Southern Taiwan [38], Okinawa, Japan [39], and the Philippines [18] in 1998, 2007, 2016, and 2017. Coral bleaching was partially curtailed by the cooling effects of typhoons in earlier years [20], as well as during our 2021 surveys, yet this was not the case in 2020; consequently, our data reflect the worst-ever bleaching event documented in Taiwan. Nearly two-thirds of all coral tissue area was bleached in September, and one-third of corals were still bleached in December 2020. As of September 2021, coral cover of the two shallow, natural reefs, HJ and HBH (~20% for each), had failed

to return to the pre-bleaching level (~40%). Although it is unclear why this is the case at these two genetically connected sites [40], one possible factor that could have thwarted recovery at HJ was an oil spill off the west coast of the Hengchun Peninsula in June 2021. Indeed, oil spills have compromised coral resilience elsewhere [41,42]. HBH has been impacted by substantial pollution, as well, albeit from a nearby recreational port and marina [43].

Bleaching was far worse at 3 vs. 7 m (78 vs. 8.5% in September 2020, respectively). This is likely not only due to the reduced light levels, but also because of the more dramatic cooling at depth brought upon by upwelling. Whether the upwelling-induced drop in seawater temperature at the 7-m site or the presumably lower light levels was more influential in the dramatically reduced bleaching incidence remains to be determined but will have important implications for modeling bleaching responses in Southern Taiwan. It is also possible that there is a greater food supply at depth; although zooplankton densities were not measured herein, they are generally higher in Nanwan Bay [27,44] than elsewhere in the Indo-Pacific [45] and could explain the relatively high resilience of these corals at thermal dosings that caused near 100% mortality elsewhere [30]. Regardless, only a single site >3 m was surveyed herein, and a far greater number of reefs must be surveyed to ultimately understand these depth-related differences in bleaching resilience [46].

Although seawater temperatures have been significantly warmer, ~1 °C warmer at OL for at least 10 years (data herein & [21]), the percentage of coral tissue area bleached (65% in September 2020), as well as the coral cover decrease (7%), was lower than, or similar to, those of the shallow natural reefs. This reef has historically demonstrated elevated resilience to marine heatwaves due to upwelling-induced thermal buffering [47], coral associations with thermotolerant Symbiodiniaceae lineages [48], and high genetic connectivity with corals of other reefs in the Western Pacific and South China Sea [49]. Whether the constituent corals possess genetic adaptations that better enable them to thrive in relatively warmer seawater [50] remains to be determined. OL is a popular site for snorkeling, SCUBA diving, free diving, and fishing [51]; it may be worth designating this area as an MPA to foster the resilience of these thermotolerant corals from additional stressors, namely physical damage from tourists [52]. Doing so could also aid in limiting overfishing impacts [53,54], as biomass of fish and other grazers will be critical to further limit the growth of the macroalgae that now dominate the benthos of OL and the other three shallow reefs (presumably driven by the increase in available substrate upon bleaching-induced coral mortality). Fish stocks have returned quickly to reefs elsewhere in Taiwan upon establishment of small marine reserves [55].

The reasons for the relatively high recovery at the *Acropora*-dominated ILI are less clear, as the temperature profiles of this artificial habitat are similar to those of HJ and HBH; furthermore, *Acropora* is well known to be amongst the most thermo-sensitive coral genera [56]. This unique reef has been the focus of study for at least 12–13 years [36], and we speculate that the constant and high flow generated by the nuclear power plant's intake turbines may act to either influence the corals' boundary layers or at least draw in high quantities of food; the latter may be evidenced by the dominance of zooplanktivorous fish such as *Dascyllus* spp. and *Chromis* spp. [57] at ILI, where access is highly regulated and oftentimes restricted due to safety concerns arising from hazardous flow regime.

Coral bleaching events provide opportunities to identify bleaching-resistant individuals within populations already exhibiting higher bleaching thresholds and have the advantage of allowing for assessment of relative performance in a natural context [58]. Complex histories of chronic and acute seawater temperature stresses are expected to trigger taxon- and location-specific responses that will ultimately lead to novel coral communities. At HJ pocilloporid coral cover remained at a consistent 11% over time (albeit with 50% of these bleached in September 2020), despite widespread bleaching of other genera. Corals of this genus have also been found to be moderately resistant to thermal stress in meta-analyses [59], and pocilloporids demonstrate high recruitment in Southern Taiwan (21–554 m$^{-2}$; [60]), possibly because they are brooders that release larvae each month [61].

As documented elsewhere [62], *Seriatopora* spp. and *Stylophora* spp. had high bleaching incidences and underwent declines in cover at HBH and OL, and *Acropora* underwent large declines in cover at ILI between April 2020 (38% of the benthos) and December 2020 (10%) before recovering to 43% by September 2021. Elsewhere, repeated bleaching events have resulted in poritid-dominant reefs [63], though montiporids, acroporids, and fire coal are currently the dominant reef-building coral species at OL, ILI, and HBH, respectively. Although poritids did increase in abundance at ILI and HBH, they do not comprise more than 3% of the benthos. Furthermore, although some genera actually increased in abundance post-bleaching (e.g., *Diploastrea* at HJ; Figure S3), it cannot be stated with certainty whether this is evidence for a phase shift on account of the emergence of thermotolerant genera replacing the more thermo-sensitive branching species that were historically most common; this could indeed be the case, but since photo-quadrat positions were not fixed, it is possible that these species were simply overlooked in earlier surveys. Perhaps the employment of a colony-by-colony fate-tracking approach in future surveys would better enable us to document the thermotolerance of less common species.

## 5. Conservation Implications and Future Directions

It has been recommended that naturally occurring climate-resilient corals be used to construct bleaching-resistant nurseries [64]. The pocilloporids at HJ and OL, as well as the acroporids at ILI, could be good candidates since they survived such an extreme bleaching event. These thermotolerant corals are now being propagated for active restoration *in situ* and *ex situ*, amidst simultaneous attempts to understand how their resilience is maintained at the physiological level [65–67]. Evidence is accumulating that relative bleaching performance is maintained following acclimatization to both aquarium [68] and distinct *in situ* conditions. Therefore, the conservation and restoration of wild corals requires not only environmental mitigation (namely a dramatic reduction in greenhouse gas emissions), but also improvements in coral transplantation and husbandry [69–71]. We have now successfully transplanted corals from HJ and elsewhere into the OL, where they have survived several months; whether they can now better withstand future marine heatwaves will be known by late 2023. Furthermore, given the dramatically reduced degree of bleaching in corals just several meters down the water column at ILO, perhaps this site could serve as a local nursery for the aforementioned thermotolerant coral genera, as well as corals that may instead be lacking in thermotolerance and so would benefit from the cooler, darker waters. More broadly, it will be critical to factor data from studies such as these featuring highly heterogeneous responses to marine heat waves in corals in close proximity, into the next generation of machine-learning [72] and other predictive models that dictate the optimal conservation strategy for a particular reef or coral species given the data in-hand [73]. In this way, we can ensure that the conservation measure(s) taken (see Table 1 for additional options.) will minimize further environmental damage and maximize odds of coral survival into the next millennium.

**Supplementary Materials:** The following are available online at https://www.mdpi.com/article/10.3390/app13095554/s1: (Appendix S1) A supplemental results section containing three supplemental figures (Figures S1–S3) and a supplemental table (Table S1) and (Appendix S2) the online supplemental data file (OSDF; zipped Excel), which includes all data found in the manuscript.

**Author Contributions:** Conceptualization: T.-Y.F.; methodology: Z.-M.Y., A.B.M. and T.-Y.F.; software: Z.-M.Y., A.B.M. and T.-Y.F.; formal analysis: Z.-M.Y., A.B.M. and T.-Y.F.; investigation: Z.-M.Y., A.B.M. and T.-Y.F.; resources: T.-Y.F.; data curation: Z.-M.Y. and A.B.M.; writing—original draft preparation: Z.-M.Y., A.B.M. and T.-Y.F.; writing—review and editing: Z.-M.Y., A.B.M. and T.-Y.F.; visualization: Z.-M.Y., A.B.M. and T.-Y.F.; supervision: T.-Y.F.; project administration: T.-Y.F.; funding acquisition: T.-Y.F. All authors have read and agreed to the published version of the manuscript.

**Funding:** This research project was funded by Taipower Company (Taiwan) under project number 061080002601 to T.Y.F.

**Institutional Review Board Statement:** This study featured only the acquisition of images of marine animals; consequently, no IRB approval or permits were required. However, special permission was obtained from TaiPower (Taiwan) to access the intake water area for their nuclear power plant (permit #1092352696 to T.-Y.F.).

**Informed Consent Statement:** Not applicable.

**Data Availability Statement:** All data found in this article have been submitted directly to MDPI as part of the online supplemental data file. Those with access to JMP® can also email A.B.M. (anderson@coralreefdiagnostics.com) to request the JMP scripts that will re-create the manuscript's figures, tables, and statistical analyses.

**Acknowledgments:** We would like to thank Chih-Jui Tan, Yan-Leng Huang, Kwok-Wai Lam, and Qun-Xue Zheng for help with SCUBA diving surveys. A.B.M. would like to acknowledge the Fulbright and MacArthur Foundations for supporting his stay in Taiwan.

**Conflicts of Interest:** The authors declare no conflict of interest.

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
