# Peer review of "Variable Responses to a Marine Heat Wave in Five Fringing Reefs of Southern Taiwan"

_applsci, doi:10.3390/app13095554_

Round 1
Reviewer 1 Report
An impressive amount of data and elaboration, but not always clearly reported and easily readable, with discussion and above all conclusions somewhat disconnected from the results presented.
Figures and tables to be improved and better adapted to the pagination of the journal.
More general comments are briefly reported below, and others in reference to the PDF file where corresponding words/sentences are marked with colours.
Can you give some ideas about the information loss induced by the photo-quadrat method you applied? Can you provide some indication or thought (and/or discuss) about the increase in the number of species found after bleaching and the heatwave events? Do you think that the discovery of further species could be linked/caused to the possible death and decay-disappearance of soft-bodied species previously covering some possibly smaller species/colonies???
It is a damage that some data are provided using different criteria (see for instance data in Fig. 4 mean tissue area bleached rather than the percentage of surveyed colonies that presented bleached tissues, as elsewhere), that makes very hard reading and comparing information!!!
It would be very relevant to know if the coral genera not reported as bleached from some localities were present but unbleached or not represented at all in those localities! Please, provide some information about this.
From the comparison of panels in figure 5, it is possible to see that some genera you consider behave in different ways in the different localities after the heatwave stress. The situation in B (the only locality with a sensible percentage of Seriatopora and a relevant quantity of Favites) is particular, with this second genus increasing sensibly in Dec. 2020. Do you know/did you examine colonies directly to understand if these genera are represented by the same species in the different sites? Could you exclude that the results you obtained are also influenced by particular filters/behaviors at species level? Because of special adaptations of particular species to particular environmental conditions and a certain vicariance between species of the same genus to colonise habitats with even slightly different conditions.
I suggest you assessing, wherever in materials and methods, that cover indications relate to live corals. Indeed, it seems strange to mention (in the description of the examined localities) that the cover of a skeletonized coral decreases because the colony skeleton would persist in the same position.
Conclusions seem to me a bit disconnected from the results and discussions.
I suggest avoiding Italic for & and normal words, such as was…
Materials and methods:
Page 2: 1-add information about depth for all localities.
Page 2: Apr., Sept.: add 2020 for completeness.
Page 2: How many images along the 30 m long transects? Or did you cover continuously the 30m transect leaving no space in between subsequent images, thus analyzing a belt 35 cm in width?? Please, specify. From the sentence in green on page 7, it seems that you covered a belt and possibly a certain overlap occurred between adjacent photos (ca. 90 per transect)? Is this correct? You must describe better how you acquired images.
Page 7: Lines 8-12: unclear: to pass from 39 to 42 categories (groups), each of these must remain separated from the others….
Page 7: Line 15: I suggest to use groups rather than taxa for these composite groupings of taxa (either belonging to the same or different genera).
Page 7: paragraph 2, line 3: and
Page 7: paragraph 2, line 5: delete full stop
Page 7: paragraph 2, lines 5-6: if you have sand, I expect, mud or silt for finer sediments.
Page 7: paragraph 2, line 7: what do you mean for: transect hardware?
Page 7: paragraph 2, lines 6-7: “Point data were converted to percentages of the total transect length” this is not fully clear to me: did you consider points? Or did you consider the lengths of the particular organisms or representatives of other categories which was present at that particular point??
Page 8: heading 2.4, line 2: somewhat wrong with the parentheses.
Page 8: °C: strange position for the degree indication…possibly an incorrect symbol has been used everywhere.
Page 10: heading 3.2, line 8: cover.
Page 10: heading 3.2, paragraph1, line 6 and paragraph 2, line 5: It is unclear to me why you give the results about bleaching as percentage of the documented coral cover in the first instance and as percentage of the total benthic cover in the second instance. This makes understanding your date more complicate!
Page 10: heading 3.2: I suggest avoiding citing repeatedly these figures after first time to simplify reading…
Page 10: heading 3.3.1: I suggest displacing this general part elsewhere, possibly at the beginning in 3.3, or to delete it at all because you systematically quote (and several times) these figures in the following!
Page 13: Discussion, line 10: something missing…
Page 15: rephrase to increase readability
Page 17: Conclusions: lines 1-2: something missing?
Page 17: Conclusions: line 11: conspecific: this is first time you use this concept, and in the entire paper only genera (and not species) have been considered. I find some inconsistency between the figures (especially Fig. 4 depicting a completely different association of this deep site in comparison to the shallower -3m deep- ones) and this statement about the occurrence of the same species. I suggest you anticipate something about before conclusions that appear at least partly disconnected from the topic addressed. And see also my comments about the reliability of the genus-level information for studies on environmental distribution.
References:
Reference 1 is missing
Figures and tables and the indication of the 5 localities: I suggest you to standardise the use of letters, font type of characters and the employment or not of the abbreviations. I prefer what you did in Figure 6 that is very clear, with only the locality reported. It is unnecessary to have both names and abbreviations.
Fig. 1. Improve accuracy in the graphics. I suggest including a general map to locate Taiwan (A), displacing Taiwan on the right side of the figure, partly superimposed on the enlargement of the studied area. Delete the locality names from the panels (presently, C, D, E, F) to avoid confusion: names are reported in the caption. Align all images also leaving the same space in between them.
In the caption, add the depth for all localities.
Figs 2, 3. Consider including the whole caption before the figure in the final pagination.
Fig. 4. It would be very important to know if the coral genera absent in these graphs for C-OL, D-ILI and E-ILO were present but unbleached or not represented at all in those localities!
Fig. 4: I suggest removing the yellow highlighted headings for each panel. This is clearly indicated by the diagrams, and is occasionally not true (panel B, second column, for instance).
Figs 4 and 5: In these images, it would be nice to have also information about the total coral cover for each locality in order to visualise easily the percentages described in the text and the real incidence of major species per locality, respectively. Furthermore, the coral cover percentage on the total investigated area on top of each column in Fig. 5 would provide a clear indication on the time changes during time at each locality!!! And this would be very important to facilitate reading!
Fig. 5. I can understand, but I do not like the indication of percentages for some species only….In any case I suggest you to write this in the caption, for instance, as follows: Values are reported when exceeding 7%.
Fig. 5, 6 and 7 must be anticipated in the text. Presently, fig. 5 is placed even after Fig. 8 is quoted in the text!
Fig. 6. Caption: Please, specify what type of cover! In order to facilitate reading. Put also letters closer or inside the panels to which they refer to!
Fig. 8. Consider revising headings for X axis. They are unclear (yellow highlighting).
Tabs 1, 2, 3.
These are incompletely visible on the PDF I received for reviewing. I think these must be reduced in size in order to be included in the final text or may be displaced in the supplementary files if they don’t fit the page size of the journal. In Table 1, I was not able to detect any letters a and b mentioned in the caption, but this could be an artifact of the incompleteness of the vision. In table 2, the meaning of the bold horizontal arrow is not reported in the caption. Table 3 caption: correct the Italic (unnecessary, missing) for two words (highlighted in yellow).
Author Response
Reviewer#1
Reviewer #1 summary: An impressive amount of data and elaboration, but not always clearly reported and easily readable, with discussion and above all conclusions somewhat disconnected from the results presented. Figures and tables to be improved and better adapted to the pagination of the journal. More general comments are briefly reported below, and others in reference to the PDF file where corresponding words/sentences are marked with colours.
Authors’ response to reviewer #1’s summary: Thank you for taking the time to critically review our article. We have done our best to address your concerns in point-by-point fashion below. Hopefully, this has resulted in a superior article worthy of publication. Regarding the overall aesthetics of the article, it is true that some figures did not render properly upon conversion to PDF, and they may appear blurry or low-resolution on certain screen (particularly small laptop screens like my own). We will work with the article processing team and provide the vector-based EPS files so that they can draft a more professional PDF for online download.
Reviewer #1’s comment #1: Can you give some ideas about the information loss induced by the photo-quadrat method you applied? Can you provide some indication or thought (and/or discuss) about the increase in the number of species found after bleaching and the heatwave events? Do you think that the discovery of further species could be linked/caused to the possible death and decay-disappearance of soft-bodied species previously covering some possibly smaller species/colonies???
Authors’ response to reviewer #1’s comment #1: This is an excellent point, and one about which we had thought amongst ourselves previously. In fact, the other lead author wanted to build a whole story around thermo-tolerant coral, phase shifts, etc. and target the likes of Nature for publication, but being the ultimate pessimist, I ultimately convinced him that, although this COULD be the case, it could also be attributed to one weakness of this method: that photo-quadrat placement was somewhat random, despite the transects being fixed. This means that we could not assess bleaching on a colony-by-colony basis. Perhaps this would have been the ideal approach (in hindsight) and there are various sentences sprinkled throughout the article that allude to this. I actually think your idea is very interesting and could very well be the case; reef “real estate” opened up for less common species post-bleaching, especially considering the fact that the most dominant species at most of the survey reefs are all known to be thermo-sensitive (acroporids, seriatoporids, montiporids in particular). Since the INCREASES in new species were generally so small (in terms of their percentage of the benthos), we had effectively glossed over this potentially interesting finding, but upon your mentioning of it, I do think it is worth mentioning, and I have now added the following sentence to the Discussion section focused on thermotolerant coral species/genera:
“Furthermore, although some genera actually increased in abundance post-bleaching (e.g., Diploastrea at HJ; Figure S3), it cannot be stated with certainty whether this is evidence for a phase shift on account of the emergence of thermotolerant genera replacing the more thermo-sensitive branching species that were historically most common; this could indeed be the case, but since photo-quadrat positions were not fixed, it is possible that these species were simply overlooked in earlier surveys. Perhaps the employment of a colony-by-colony “fate-tracking” approach in future surveys would better enable us to document the thermotolerance of less common species with greater confidence.”
Reviewer #1 comment #2: It is a damage that some data are provided using different criteria (see for instance data in Fig. 4 mean tissue area bleached rather than the percentage of surveyed colonies that presented bleached tissues, as elsewhere), that makes very hard reading and comparing information!!!
Author response to reviewer #1’s comment #2: I can see your point, and we struggled with this ourselves, but the reason is because these data, while seemingly similar, actually tell very different stories. In earlier tables and figures, we focus on the reef areas as a whole and report bleaching across all corals, or the most common ones. Figure 4 is basically a higher resolution view and is meant more to show differences in thermotolerance. If you quickly skim through the figures without reading the text in between them, you could certainly become lost! The other reason is because throughout the entire manuscript, we focus on “area;” the percentage of coral surface area bleached. The reason is because simply stating whether or not a colony had bleached (the traditional way) is far lower resolution. If a colony was 1% bleached, it could be scored as “bleached.” This would then over-estimate the severity of bleaching. By reporting bleaching instead as percent of tissue area bleached, the magnitude and severity of bleaching is more accurately presented.
Reviewer #1 comment #3: It would be very relevant to know if the coral genera not reported as bleached from some localities were present but unbleached or not represented at all in those localities! Please, provide some information about this.
Author response to reviewer #1’s comment #3: If a species was not found on a reef, we would not report its bleaching one way or the other. To put it another way, all corals photographed would be scored as bleached or unbleached. We did not assume that corals that were NOT imaged were healthy or bleached. This is one reason why we did not draw too much meaning from the emergence of new genera (raised in your comment above); they weren’t necessarily thermo-tolerant “survivors.” They may simply have been overlooked!
Reviewer #1 comment #4: From the comparison of panels in figure 5, it is possible to see that some genera you consider behave in different ways in the different localities after the heatwave stress. The situation in B (the only locality with a sensible percentage of Seriatopora and a relevant quantity of Favites) is particular, with this second genus increasing sensibly in Dec. 2020. Do you know/did you examine colonies directly to understand if these genera are represented by the same species in the different sites? Could you exclude that the results you obtained are also influenced by particular filters/behaviors at species level? Because of special adaptations of particular species to particular environmental conditions and a certain vicariance between species of the same genus to colonise habitats with even slightly different conditions.
Author response to reviewer #1’s comment #4: This is an excellent point, yet, despite two of the three authors being local experts on Taiwanese coral taxonomy (myself excluded), the diversity of Southern Taiwan is simply too high and too complex to go beyond the genus level of identification in all but select cases. It could very well be the case that one species of Favites behaves completely different from another (e.g., one being significantly more thermo-tolerant), yet our approach’s resolution was too low. As the artificial intelligence becomes better trained in the future (a surety), we anticipate that it will be able to accurately resolve species-level differences from even inferior-quality images, which will then let us craft a more detailed picture in terms of the changing community dynamics.
Reviewer #1 comment #5: I suggest you assessing, wherever in materials and methods, that cover indications relate to live corals. Indeed, it seems strange to mention (in the description of the examined localities) that the cover of a skeletonized coral decreases because the colony skeleton would persist in the same position.
Author response to reviewer #1’s comment #5: We did include a category for “dead hard corals” but these data were binned with the abiotic parameters. In reality most dead hard coral would quickly be covered by turf algae (and scored as such). But you are correct in noting that it should be emphasized that only live coral were considered in the majority of data presented, and I have now revised the materials and methods to
emphasize this. The exception might be those corals that had bleached to the extent of effective death, yet they were not yet being overgrown with algae. In these cases, they might be considered “bleached” when one could argue that they are technically “dead hard coral.”
Reviewer #1 comment #6: Conclusions seem to me a bit disconnected from the results and discussions.
Author response to reviewer #1’s comment #6: This is a good point and yet another one where the other lead author and I disagreed. However, the second reviewer actually wanted us to talk MORE about implications of the work (i.e., the focus of the Conclusions) and so we have generally left it as is).
Reviewer #1 minor comment #1: I suggest avoiding Italic for & and normal words, such as was…
Author response to reviewer #1’s minor comment #1: We had used italics to provide emphasis, though I recall now that this is actually against MDPI’s policy, so all such instances will be changed back to normal font during proofing.
Materials and methods:
Reviewer #1 minor comment #2: Page 2: 1-add information about depth for all localities.
Author response to reviewer #1’s minor comment #2: This is a good catch, as the depths had been mentioned only for 2-3 of the sites, not all of them.
Reviewer #1 minor comment #3: Page 2: Apr., Sept.: add 2020 for completeness.
Author response to reviewer #1’s minor comment #3: The suggested change has been made.
Reviewer #1 minor comment #4: Page 2: How many images along the 30 m long transects? Or did you cover continuously the 30m transect leaving no space in between subsequent images, thus analyzing a belt 35 cm in width?? Please, specify. From the sentence in green on page 7, it seems that you covered a belt and possibly a certain overlap occurred between adjacent photos (ca. 90 per transect)? Is this correct? You must describe better how you acquired images.
Author response to reviewer #1’s minor comment #4: This is a good point, and it will be critical to ensure that the reader understands the approach. I have now revised this section to emphasize the fact that it was, as you point out, a belt: “To examine changes in the benthic and coral community composition before (Apr. 2020), during (Sept. 2020), and after (Dec. 2020, Apr. 2021, & Sept. 2021) the 2020 bleaching event, 35 × 35 cm photo-quadrat images were taken within the same triplicate 30-m transects at each site (~85-86/transect/survey time).” In other words: 30 m = 3,000 cm / 35 cm quadrat = ~85 images/transect.
Reviewer #1 minor comment #5: Page 7: Lines 8-12: unclear: to pass from 39 to 42 categories (groups), each of these must remain separated from the others….
Author response to reviewer #1 minor comment #5: Prior to that sentence, there were 39 stony coral categories. We then added three non-scleractinian taxa, which resulted in a new total of 42 reef-building coral categories. I have reworded the sentence in question for clarity.
Reviewer #1 minor comment #6: Page 7: Line 15: I suggest to use groups rather than taxa for these composite groupings of taxa (either belonging to the same or different genera).
Author response to reviewer #1’s minor comment #6: This is a good point since, in many cases, they are NOT genera and include non-taxonomic terms like “unidentified coral.” So we now use “group” or “groupings” (or “categories”).
Reviewer #1 minor comment # 7: Page 7: paragraph 2, line 3: and
Author response to reviewer #1’s minor comment #7: In this case, we are giving all of the options, so “or” is the correct word.
Reviewer #1 minor comment #8: Page 7: paragraph 2, line 5: delete full stop
Author’s response to reviewer #1’s minor comment #8: When there is a full sentence within parentheses, the period is needed.
Reviewer #1’s minor comment #9: Page 7: paragraph 2, lines 5-6: if you have sand, I expect, mud or silt for finer sediments.
Author’s response to reviewer #1’s minor comment #9: This is a good point, and I have now mentioned this in parentheses.
Reviewer #1’s minor comment #10: Page 7: paragraph 2, line 7: what do you mean for: transect hardware?
Author response to reviewer #1’s minor comment #10: This corresponds to things like the rebar or the transect spool. I have now mentioned this in parentheses.
Reviewer #1’s minor comment #11: Page 7: paragraph 2, lines 6-7: “Point data were converted to percentages of the total transect length” this is not fully clear to me: did you consider points? Or did you consider the lengths of the particular organisms or representatives of other categories which was present at that particular point??
Author response to reviewer #1’s minor comment #11: It is actually similar to what happens in CPCE, only that the AI does most of the work. The point data actually get converted into areas within the quadrat, and so what is ultimately reported is the percent cover over a 30 m x 0.035 m (~10 m2) belt. I have now tried to better explain this in the text. In other words, you have 90 images for each transect. Each image represents a 35 x 35 cm area. The AI calculates the percent cover of the various groups within each 0.1 m2 area. You then average across all 90 images to get a transect mean, and that is what is reported. Note that this is different from a point-intercept survey, in which you would be ADDING the cover values and dividing by the total length of the transect.
Reviewer #1’s minor comment #12: Page 8: heading 2.4, line 2: somewhat wrong with the parentheses.
Author response to reviewer #’1 minor comment #12: We did not detect an issue with the parentheses but will doublecheck during the PDF conversion to make sure.
Reviewer #’1 minor comment #13: Page 8: °C: strange position for the degree indication…possibly an incorrect symbol has been used everywhere.
Author response to reviewer #1’s minor comment #13: This is a good catch, and I wonder if it was just a small font. I have now changed it to be correctly positioned.
Reviewer #1’s minor comment # 14: Page 10: heading 3.2, line 8: cover.
Authors response to reviewer #1’s minor comment #14: Another good catch! The suggested change has been made.
Reviewer #1’s minor comment #15: Page 10: heading 3.2, paragraph1, line 6 and paragraph 2, line 5: It is unclear to me why you give the results about bleaching as percentage of the documented coral cover in the first instance and as percentage of the total benthic cover in the second instance. This makes understanding your date more complicate!
Author response to reviewer #1’s minor comment #15: This is because we wanted to first emphasize that the reefs have very different responses and coral cover. When we specifically discuss bleaching, we instead focus ONLY on the coral cover areas since it doesn’t make sense to assess bleaching against a backdrop of algal cover, hard bottom, etc. When we want to look at ecosystem-scale changes, we conducted the more traditional, “whole benthos” analysis. When we wanted to focus on bleaching dynamics, we looked at the coral data only (i.e., Figure 4). This is why bleaching data are presented in two different ways. Figures 3 and 4, then, are not meant to be directly comparable, though I can definitely see the frustration in this.
Reviewer #1 minor comment #16: Page 10: heading 3.2: I suggest avoiding citing repeatedly these figures after first time to simplify reading…
Author response to reviewer #1’s minor comment #16: This is a good suggestion, and we have removed several of the in-text figure citations in this section.
Reviewer #1 minor comment #17: Page 10: heading 3.3.1: I suggest displacing this general part elsewhere, possibly at the beginning in 3.3, or to delete it at all because you systematically quote (and several times) these figures in the following!
Author response to reviewer#1’s minor comment #17: This sounds repetitive, but it’s because we are specifically referencing the multivariate data analysis here. However, I have taken your suggestion to move the introductory sentence to the preceding suggestion to better give context and frame the site-by-site analyses to follow.
Reviewer #1’s minor comment #18: Page 13: Discussion, line 10: something missing…
Author response to reviewer#1’s minor comment #18: You are correct. There was an error in this sentence. It has been corrected to read : “Although it is unclear why this is the case at these two genetically connected sites [40], one possible factor that could have thwarted recovery at HJ was an oil spill off the west coast of the Hengchun Peninsula in June 2021.”
Reviewer #1’s minor comment #19: Page 15: rephrase to increase readability
Author response to reviewer #1’s minor comment #19: I was unsure which exact section you were referring to, but I have reread the page in question and revised where necessary. Hopefully, our overall message is now clear.
Reviewer #1 minor comment #20: Page 17: Conclusions: lines 1-2: something missing?
Authors response to reviewer #1’s minor comment #20: I think this issue (which you raised above) stems from “Conclusions” being the wrong word; it should be “implications” or “future directions.” I am not sure if Applied Sciences allows for this, but I have changed it to “Conservation implications and future directions” for now.
Reviewer #1 minor comment #21: Page 17: Conclusions: line 11: conspecific: this is first time you use this concept, and in the entire paper only genera (and not species) have been considered. I find some inconsistency between the figures (especially Fig. 4 depicting a completely different association of this deep site in comparison to the shallower -3m deep- ones) and this statement about the occurrence of the same species. I suggest you anticipate something about before conclusions that appear at least partly disconnected from the topic addressed. And see also my comments about the reliability of the genus-level information for studies on environmental distribution.
Author response to reviewer #1’s minor comment #21: Firstly, we have removed the word “conspecific” since, as you correctly noted, we did not undertake a species-specific analysis. Secondly, Figure 3 shows that unbleached coral cover ranged from 38-52% at the deeper site. In contrast, unbleached coral cover dropped to 17-18% from ~40% at two of the sites, recovering at the other two. Figure 4 shows that the overall bleaching percentage at the deep site was 8.5% vs. 65-97% at the other sites. We take this to mean that bleaching severity was less at the deep site.
References:
Reviewer #1 minor comment #22: Reference 1 is missing
Author’s response to reviewer #1’s minor comment #22: This is not the case in our document. Perhaps there was an issue with the PDF conversion. I will try to do a better job of checking the PDF proofs next time, as evidently there were issues.
Reviewer #1 minor comment #23: Figures and tables and the indication of the 5 localities: I suggest you to standardise the use of letters, font type of characters and the employment or not of the abbreviations. I prefer what you did in Figure 6 that is very clear, with only the locality reported. It is unnecessary to have both names and abbreviations.
Author response to reviewer #1’s minor comment #24: We have taken your suggestion and removed the site abbreviations except for in Table 1.
Reviewer #1 minor comment #24: Fig. 1. Improve accuracy in the graphics. I suggest including a general map to locate Taiwan (A), displacing Taiwan on the right side of the figure, partly superimposed on the enlargement of the studied area. Delete the locality names from the panels (presently, C, D, E, F) to avoid confusion: names are reported in the caption. Align all images also leaving the same space in between them.
In the caption, add the depth for all localities.
Author response to reviewer #1’s minor comment #24: This is a good suggestion, and we have remade Figure 1 accordingly. We re-sized and re-centered the images and then removed the site names and abbreviations since, as you mentioned, they are stated in the caption.
Reviewer #1 minor comment #25: Figs 2, 3. Consider including the whole caption before the figure in the final pagination.
Author response to reviewer #1’s minor comment #25: I am almost certain this will be the case, but I will double-check with MDPI at a later stage in the process.
Reviewer #1 minor comment #26: Fig. 4. It would be very important to know if the coral genera absent in these graphs for C-OL, D-ILI and E-ILO were present but unbleached or not represented at all in those localities!
Author response to reviewer #1 minor comment #26. This is a good point, and now I understand why you raised this point above. In the case of Lobophyllia, it actually did NOT bleach. I have added “0’s” in these instances. When a coral was simply “not found,” I have a “NF” in the respective spot. This is interesting in fact because Lobophyllia was the MOST bleaching-susceptible at other sites. It is a shame, then, that, as you pointed out, we did not conduct a species-by-species analysis. I feel certain that some Lobophyllia species must be MUCH stronger than others based on this!
Reviewer #1 minor comment # 27: Fig. 4: I suggest removing the yellow highlighted headings for each panel. This is clearly indicated by the diagrams, and is occasionally not true (panel B, second column, for instance).
Author response to reviewer #1 minor comment #27: We have removed the grey shading around the panel headings.
Reviewer #1 minor comment #28: Figs 4 and 5: In these images, it would be nice to have also information about the total coral cover for each locality in order to visualise easily the percentages described in the text and the real incidence of major species per locality, respectively. Furthermore, the coral cover percentage on the total investigated area on top of each column in Fig. 5 would provide a clear indication on the time changes during time at each locality!!! And this would be very important to facilitate reading!
Author response to reviewer #1’s minor comment #28: We have now added the data from Figure 3 into Figure 5 to show unbleached coral cover over time.
Reviewer #1 minor comment # 29: Fig. 5. I can understand, but I do not like the indication of percentages for some species only….In any case I suggest you to write this in the caption, for instance, as follows: Values are reported when exceeding 7%.
Author response to reviewer #1’s comment #29: This is because different species were dominant at different sites. It is possible to be amongst the top five species in the region but not be present at all in a particular site; therefore, some values will be under 7%. The main point is to show trends in the most dominant species.
Reviewer #1 minor comment #30: Fig. 5, 6 and 7 must be anticipated in the text. Presently, fig. 5 is placed even after Fig. 8 is quoted in the text!
Author response to reviewer #1’s minor comment #30: This has to do with the fact that they are referenced in Table 1 and so they are technically in order, even if they aren’t discussed until later. This is a strange requirement of MDPI, and so I may later opt to remove their citations within Table 1 for this very reason.
Reviewer #1 minor comment #31: Fig. 6. Caption: Please, specify what type of cover! In order to facilitate reading. Put also letters closer or inside the panels to which they refer to!
Author response to reviewer #1 minor comment #31: The type of cover is on the y-axis. I have moved the panel labels closer to the panels themselves.
Reviewer #1 minor comment #32: Fig. 8. Consider revising headings for X axis. They are unclear (yellow highlighting).
Author response to reviewer #1’s comment #32: They are in black on my screen. Perhaps there was an issue with the MDPI PDF conversion. I will check.
Tabs 1, 2, 3.
Reviewer #1 minor comment #33: These are incompletely visible on the PDF I received for reviewing. I think these must be reduced in size in order to be included in the final text or may be displaced in the supplementary files if they don’t fit the page size of the journal. In Table 1, I was not able to detect any letters a and b mentioned in the caption, but this could be an artifact of the incompleteness of the vision. In table 2, the meaning of the bold horizontal arrow is not reported in the caption. Table 3 caption: correct the Italic (unnecessary, missing) for two words (highlighted in yellow).
Author response to reviewer #1’s minor comment #33: Thank you for pointing this out, and I apologize for the poor PDF conversion. It looked fine on my screen, but perhaps some changes were made by the editorial office. The superscripts are indeed there in Table 1. In table 2, the bold arrow simply denotes the passing of time. For Table 3, I have removed the italics from the headings, except for “Approx./Exact F”, which should be in italics. I also left genus names in italics.
Reviewer 2 Report
This paper provides an analysis of the effects of the recent 2020 heat wave on coral bleaching at different case study sites in southern Taiwan.
The paper has an impressive array of analyses and data, much of which is included in the figures and supplementary material. But until I reached the conclusion I was left with a sense of “so what”? The conclusion provided some good suggestions of how restoration of coral reefs could be achieved. I did think though that more could be made of the lessons learnt from the study. The publication is called, “applied sciences”. How can this research be applied to provide useful management advice? For example, the one of the case study sites was affected by a nuclear power plant, yet there were no recommendations in the paper for how to deal with this.
Other comments
The paper is generally well written, but there were a number of small editorial issues that need to be corrected (words omitted and the like).
The formatting of a number of the figures and tables were poor in my version of the manuscript. It seems for example that a landscape page layout was used in some cases in the original diagrams, yet was now in portrait format, thereby truncating the figures and tables.
Author Response
Reviewer 2
This paper provides an analysis of the effects of the recent 2020 heat wave on coral bleaching at different case study sites in southern Taiwan.
Reviewer #2 summary: The paper has an impressive array of analyses and data, much of which is included in the figures and supplementary material. But until I reached the conclusion I was left with a sense of “so what”? The conclusion provided some good suggestions of how restoration of coral reefs could be achieved. I did think though that more could be made of the lessons learnt from the study. The publication is called, “applied sciences”. How can this research be applied to provide useful management advice? For example, the one of the case study sites was affected by a nuclear power plant, yet there were no recommendations in the paper for how to deal with this.
Author response to reviewer #2’s summary: You are correct, and frankly, we were somewhat deflated by this massive undertaking because our initial hypothesis was that either a) upwelling would save corals, b) that corals of the nuclear power plant outlet would have superior thermotolerance across the board, and/or c) that the artificial reefs in the area might serve as refugia relative to the natural sites. In actuality, the picture was much more muddled, and bleaching responses were heterogeneous across taxa, space, and time. We can say that the remaining corals at the Outlet have either adapted or acclimatized to abnormally elevated temperatures, making them potentially suitable candidates for restoration projects. In fact, we are developing a coral nursery at this site for just this purpose, and corals transplanted from elsewhere survived transplantation and subsequent inoculation in this overall warmer habitat. I think it will be important to mention this, so I have expanded the Conclusions section to try and provide a more compelling story in terms of what this means for coral conservation. You will also see some conservation implications in Table 1.
Other comments
Reviewer #2 minor comment #1: The paper is generally well written, but there were a number of small editorial issues that need to be corrected (words omitted and the like).
Author response to reviewer #2’s comment #1: Reviewer #1 also identified a few typographical errors, which we have now corrected. Upon multiple readings, we hope that it is now error-free.
Reviewer #2 comment #2: The formatting of a number of the figures and tables were poor in my version of the manuscript. It seems for example that a landscape page layout was used in some cases in the original diagrams, yet was now in portrait format, thereby truncating the figures and tables.
Author’s response to reviewer #2 comment #2: This was clearly the case since both reviewers reported the same thing. In my PDF copy of the manuscript, the tables and figures were rendered appropriately. The tables are massive and do need to be presented horizontally to look good. We will ensure that the next set of proofs look better and apologize for the inconvenience.